# A Review on Recent Advancements of Ni-NiO Nanocomposite as an Anode for High-Performance Lithium-Ion Battery

**DOI:** 10.3390/nano12172930

**Published:** 2022-08-25

**Authors:** Safina-E-Tahura Siddiqui, Md. Arafat Rahman, Jin-Hyuk Kim, Sazzad Bin Sharif, Sourav Paul

**Affiliations:** 1Department of Mechanical Engineering, Chittagong University of Engineering and Technology, Chittagong 4349, Bangladesh; 2Clean Energy R&D Department, Korea Institute of Industrial Technology, 89 Yangdaegiro-gil, Ip-jang-myeon, Seobuk-gu, Cheonan-si 31056, Chungcheongnam-do, Korea; 3Department of Mechanical Engineering, International University of Business Agriculture and Technology, Dhaka 1230, Bangladesh

**Keywords:** lithium-ion battery, nanocomposite, nickel oxide, anode

## Abstract

Recently, lithium-ion batteries (LIBs) have been widely employed in automobiles, mining operations, space applications, marine vessels and submarines, and defense or military applications. As an anode, commercial carbon or carbon-based materials have some critical issues such as insufficient charge capacity and power density, low working voltage, deadweight formation, short-circuiting tendency initiated from dendrite formation, device warming up, etc., which have led to a search for carbon alternatives. Transition metal oxides (TMOs) such as NiO as an anode can be used as a substitute for carbon material. However, NiO has some limitations such as low coulombic efficiency, low cycle stability, and poor ionic conductivity. These limitations can be overcome through the use of different nanostructures. This present study reviews the integration of the electrochemical performance of binder involved nanocomposite of NiO as an anode of a LIB. This review article aims to epitomize the synthesis and characterization parameters such as specific discharge/charge capacity, cycle stability, rate performance, and cycle ability of a nanocomposite anode. An overview of possible future advances in NiO nanocomposites is also proposed.

## 1. Introduction

The International Energy Agency (IEA) reports that total global power consumption was 22,315 billion kWh in the year 2018, which showed a rise of 4% over 2017 [1]. In order to meet the fast-growing demand for power worldwide, electricity generation reached an estimated 27,644 billion kWh in 2019 and approximately 63% of that electricity was generated from fossil fuels [1,2]. Though there was a 2.5% decrease in demand for power in the first few months of 2020 because of the COVID-19 lockdown, the anticipation of the U.S. Energy Administration (EIA) is that the demand will rocket from 34,000 to 42,000 billion kWh/year by 2040 [1]. This inexorable growing demand for fossil fuels and other traditional energy sources poses a significant threat to the environment by raising the level of CO_2_ emissions. It is reported that about one-fifth of CO_2_, one-third of CFCs, and half of NO_x_ in the environment are associated with the burning of fossil fuels [3]. To alleviate the repercussion of such emissions, the responsible authorities have undertaken recovery policies to decrease emissions to half by 2030 and to attain net-zero emissions by 2050 to accomplish the 1.5 Celsius goal (according to the Paris Agreement) [4].

Therefore, it is high time to consider alternative sources such as clean and renewable energy as the prime source of energy. However, renewable sources are usually remittent and periodical. The necessity of uninterrupted electricity supply for the household and the industrial sector requires a continuous power supply to function properly. Therefore, numerous investigations on Energy Storage Systems (ESSs) have been carried out to fulfill the increasing electricity demand and eradicate environmental pollution. One of the best solutions to this problem would be a battery that can store energy and can deliver according to demand. Required attributes include quick response-ability and discharge-ability, unlike non-renewable fossil fuels. Moreover, battery energy storage offers high energy density along with large storage capability [5]. A comparative study of different ESSs with respect to specific power or power density and specific energy requirements is represented in Figure 1. It is noted that a battery can be the best feasible choice as it belongs to a relatively high energy density group.

Among all the battery technologies, the LIB is the state-of-the-art in battery technology as it offers high gravimetric and volumetric energy density which enables it to store energy densely in a compact space. The lighter weight with commendable energy density has made LIB suitable for electric vehicles (EV), hybrid electric vehicles (HEV), and plug-in hybrid electric vehicles (PHEV). Moreover, the low maintenance cost, eco-friendly nature, low self-discharge, easy manufacturing, higher cell voltage, no priming requirement, and no memory effect are some other considerable attributes of LIB technology [6,7,8]. In addition, LIB shows better performance regarding energy densities compared to other commercial rechargeable batteries as shown in Figure 2 [9].

Back in 1970, chemist John Goodenough along with his co-researchers Phil Wiseman, Koichi Mizushima, and Phil Jones from Oxford University introduced the LIB packs and published their findings in 1980 [10]. In the early 1990s, meanwhile, Goodenough came up with a high energy density and high-performance layered cathode material (LiCoO_2_) [9] followed by spinel manganese as a low-cost cathode material in the year 1983 [11]. However, the application of layered cathode materials with their counterpart was limited due to the lack of low-risk anode material. This was taken into account by Besenhard et al. [12], Yazami et al. [13], and Basu et al. [14] in the late 1970s and early 1980s. They demonstrated a layered graphite structure that could reversibly entrap lithium-ions through intercalation and de-intercalation. Yohsino et al. [15] constructed a highly stable sample battery in air consisting of carbon as anode with its counterpart LiCoO_2_ as a cathode. The favorable design of this battery was later commercialized by a Japanese company (Sony) and went to large-scale production in 1990. Since 1991, carbon-based materials such as graphite are employed as anodes in commercially available LIB due to cost-efficiency, good reversibility, good electronic conductivity, high operational potential, and low volume variations during Li^+^ intercalation/de-intercalation, and high Li^+^ chemical diffusion coefficient. These factors grabbed the attention of storage energy seekers and thus the demand for LIB is growing rapidly in the field of electrically powered vehicles, portable electronics, and uninterrupted energy supply (UES) [16]. Figure 3 represents the growing demand for LIB over two decades.

Nevertheless, long-range electric vehicles experience inadequate power density, as 150–265 Wh/kg energy density is not sufficient [17]. In addition, the limited capacity of graphite anode (372 mAh g^−1^) is inadequate for long-run EVs, HEVs, and other portable devices as they require high energy density as well as high power density [18,19]. Moreover, significant structural collapse and exfoliation during cycling, strong polarization at a fast charge/discharge current rate, formation of lithium dendrites on the graphite surface, low working voltage (0.1 V vs. Li/Li^+^), low working temperature, and heating of the device are the major concerns [20]. Furthermore, the conventional electrode of commercial LIB was constructed through a slurry coating procedure where bulk carbon, Super P and PVDF amalgamated together and coated onto current collector. During the electrochemical operation, the extensive volume change in the active material causes the material to deflect from the current collector. This distorted active material falls into and mixes with the electrolyte as dead weight thus diminishing the capacity of the cell. Consequently, there is an urgency for designing and developing suitable anode material, an improvement in existing materials, and substituting graphite which can hold plenty of lithium-ion within. These may improve the structural stability, and the capacity of the battery for applications such as electrification of vehicles, and grid-scale energy storage. Undoubtedly, nanocomposite materials are currently of interest as LIB anode, as they incorporate nanosized particles within the matrix of standard material possessing high surface area, high surface-to-volume ratio, and electrochemically stable structure. Moreover, nanocomposites demonstrate dramatically improved mechanical strength, structural stability, toughness, and thermal or electrical conductivity. These characteristics pave the way for new reaction sites, shorten Li^+^ transportation distance, improve the kinetics of Li^+^, and enhance specific capacity and cycle stability, which makes them feasible anode material. It is noted that the nanocomposites of NiO have gained particular attention as LIB anode whereas the Ni-NiO nanocomposite is the most promising one. In this review article, we try to elucidate the issues regarding conventional anodes, and how to overcome the drawbacks by introducing and analyzing the performance of different materials as anodes. Moreover, we briefly discuss different nanocomposite materials and particularly highlight binder involved Ni-NiO nanocomposite materials as a LIB anode. Furthermore, the advantages and disadvantages of Ni-NiO nanocomposite are elucidated.

Hence, this review is summarized as follows: Firstly, we briefly describe the insertion/extraction mechanism of LIB. Subsequently, we demonstrate a brief overview of different anode materials based on reaction mechanisms and their nanocomposites including major challenges. Thereafter, we focus on the performance of NiO nanocomposites with carbon, graphene nanosheets, carbon nanotubes and reduced graphene oxides, and on how Ni-NiO nanocomposite outweighs the disadvantages of the abovementioned nanocomposites. Finally, we offer some concluding remarks on important issues to consider and suggest future research directions.

## 2. Working of the Lithium-Ion Battery

The working principle of LIB is based on reversible intercalation/de-intercalation of Li^+^ into electrodes. LIB acts as an electrochemical cell as a result of a difference in potential between the electrodes. An equilibrium condition is achieved between electrodes through the oxidation in the anode and the reduction in the cathode. Mainly, the cathode ascertains the battery voltage and capacity and is a major active host/source of the Li-ions. The anode permits electrical current to pass through an outer circuit during discharge while on the other hand it stores Li^+^ during charging of the battery. The anode is separated from the cathode by a microporous membrane of polymer which is permeable only to the lithium ions.

During the charging process (Figure 4a), lithium ions pass through the electrolyte from the cathode (LiCoO_2_, LiMnO_4_, LiFePO_4_) to the anode (graphite). Electrons also travel from the positive to the negative electrode by taking a longer route around the outside circuit. These electrons and ions combine in the cathode which is responsible for the deposition of lithium. There will be no further deposition of lithium due to ion flow if the battery is completely charged.

The electrochemical reaction of LiCoO_2_ cathode and graphite anode is as follows:(1)$$\mathrm{Anode}\;\mathrm{reaction}:\;C+{\mathrm{xLi}}^{+}+{\mathrm{xe}}^{-}↔{\mathrm{Li}}_{x}C_{6}$$
(2)$$\mathrm{Cathode}\;\mathrm{reaction}:\;{\mathrm{LiCoO}}_{2}↔{\mathrm{Li}}_{1-x}{\mathrm{CoO}}_{2}+{\mathrm{xLi}}^{+}+{\mathrm{xe}}^{-}$$

Discharging the battery shown in (Figure 4b) is nothing but the back flow of ions through the electrolyte from the anode to the cathode. The direction of electron flow is the same but through the outer circuit and powering the connected device. During depletion of the battery, ions and electrons combine at the cathode and lithium deposition occurs. The battery will be fully discharged when there is no remaining ion to move towards to the cathode, meaning recharging is required.

LIB is an electrochemical energy storage system where chemical energy is converted into electrical energy or vice versa by the shuttle chair mechanism of lithium ions. The electrodes are generally a complex system, including Cu/Al foil as a current collector for the anode and a cathode that comprises active materials such as graphite and LiCoO_2_, respectively.

## 3. Anode Materials

Essential criteria for the selection of an appropriate intercalation-based anode material include: low cost, good durability, lightweight, high specific capacity, low irreversible loss in the initial cycle, high coulombic efficiency, fast diffusion of Li^+^ within the anode, high ionic and electronic conductivity, minimum structural changes during shuttle chair mechanism, voltage matching with the preferred cathode, stable Solid Electrolyte Interface (SEI) layer formation capability upon cycling as well as a discharge voltage plateau within 2 V. Conventionally, graphite is used as anode for LIB in spite of having some difficulties. To replace graphite for enhanced electrochemical performance a material must possess some requirements such as: (i) lower atomic weight, low density, huge quantities of lithium per unit area; (ii) cyclable, stable, high specific capacity; (iii) potential vicinity to lithium metal against high voltage cathode, high operating voltage; (iv) chemically inert with electrolyte salts or solvents and insoluble in electrolyte solvents; (v) low price, environment friendly, safe, and with good electrical conductivity [21,22].

These requirements led to numerous investigations especially on silicon [23,24], tin-based materials [25,26], composite alloys [27,28,29], metal oxides [21], metal nitrides [30,31], and sulfides [32,33,34] along with materials having relatively higher specific capacities than graphite. In addition, some other less commercialized materials such as phosphorus (2596 mAh g^−1^), antimony (660 mAh g^−1^), germanium (1600 mAh g^−1^), transition metal oxides (TMOs), and layered transition metal oxides are promising candidates for high capacity anode material [35]. Based on lithium insertion/extraction mechanisms, these metal oxides (MOs) are subdivided into three categories (Figure 5): (a) Alloy reaction mechanism—includes using compound alloys to create a lithium insertion host which maintains a strong structural bond with intermediate and lithiated phases to minimize the volume expansion during the reaction. Si, Sn, Al, Bi, Ge, SnO_2_, etc. follow this mechanism. (b) Insertion/extraction reaction mechanism—includes the insertion and extraction of lithium in and out from the lattice structure of the transition metal oxide. TiO_2_, Li_4_Ti_5_O_12_, etc. follow this mechanism. (c) Conversion reaction mechanism includes the formation and decomposition of Li_2_O with concomitant reduction and oxidation of metal nanoparticles. As in conversion-based materials, the change in structure and composition of electrode materials occurs, so a good conversion reaction-based electrode material must have the competency to regain its initial structure and composition for stable performance during the reverse process. Typically, transition metal oxides (Mn_x_O_y_, NiO, Fe_x_O_y_, CuO, Cu_2_O, MoO_2,_ etc.) and M_x_X_y_, where X = Sulphides (S), Phosphides (P), Nitrides (N), follow this mechanism [36].

At present, nanocomposite materials are attracting extensive research interest as anodes for LIB. In general, nanocomposites are multiphasic materials in which a phase must possess one, two, or three dimensions below 100 nm, or nanoscale distances exist between the phases. Nanocomposites can be formed through amalgamating inorganic nanoclusters, metals, oxides, and semiconductors with different metallic compounds. Nanocomposites have emerged as favorable alternatives to bulk engineering materials that have certain limitations. The presence of nanoparticle phases within the composite structure facilitates catalytic activity and substantial improvements in mechanical properties, flexibility, thermal stability, and improved electrical conductivity [37,38]. It is noted that some nanocomposite materials are 1000 times tougher as compared to their bulk counterparts. These unique characteristics of nanocomposites mainly arise from small-sized particles, high surface area, and possible interfacial interaction between constituent phases. Consequently, these promising characteristics lead nanocomposites to be employed as propitious anode material for LIB.

The constituents of the composite anode are an active material, a conductive material (improves electrode conductivity), and a binding agent (improves adhesion of active material with current collector and cohesion between the active material particles). The active electrode material must possess a high specific capacity and ability to store an ample amount of lithium-ion and retain the structural integrity in continuous cycling. Due to higher theoretical capacity as compared to graphite, nanocomposites of silicon (theoretical capacity 4200 mAh g^−1^), titanium oxide (theoretical capacity 336 mAh g^−1^), and transition metal oxides (500–1000 mAh g^−1^) have established themselves as promising anode active material. However, bulk or micrometer-sized Si particles suffer from poor electrical conductivity, poor cycle stability, low Li^+^ diffusion rate at room temperature, and extensive volume expansion up to 300–400%. In addition, active electrode material gets pulverized as well as loses connection with the electrode during cycling [39,40,41,42,43]. Kim et al. [44] synthesized silicon nanoparticles of 5–20 nm. Among them, as-synthesized electrodes fabricated with 10 nm silicon nanoparticle exhibited maximum 81% capacity retentions at a 0.20 C rate after 40 cycles followed by 71% and 67% for 5 and 20 nm, respectively. After 40 consecutive cycles, the Si nanoparticles aggregated severely and exhibited mixed amorphous and crystalline phases, which leads to capacity fade. Chen et al. [45] compared commercial Si nanoparticles and 2D mesoporous Si and revealed that the commercial Si nanoparticle electrode exhibits serious capacity fade after 10 cycles, which may be attributed to the extensive volume variation during the cycling process. However, the 2D mesoporous Si exhibits enhanced performance. Furthermore, composites of Si with carbon or other material can offer better performance as shown in Table 1.

Another option could be titanium oxide (TiO_2_) which possesses a stable structure during lithium inclusion/exclusion, no electrolyte decomposition issue or short circuit induced swelling during cycling. However, it has a lesser theoretical capacity (336 mAh g^−1^) than graphite, [52,53,54,55]. Therefore, its practical implications as anode for LIB were impeded due to low ion and electrical conductivity that leads to limited capacity and poor cycle performance [56,57,58]. A report on a TiO_2_-based anode revealed that the cell employing TiO_2_ experienced the inability to attain a high operating potential due to its high discharge plateau [59,60]. Due to the poor rate capability of TiO_2_ full cells, LIB experiences low power density. Borghols et al. [61] studied the lithium intercalation mechanism of anatase-TiO_2_ and found that the material suffered poor transportation of Li^+^, and therefore, further insertion of lithium could only be attained in the surface layer. Particularly, reducing the size of the particle facilitates lithium insertion/de-insertion, enhances Li^+^ diffusion, and shortens the length of the charge path. However, while TiO_2_ has high cyclability and stability, it cannot be a feasible choice where high specific capacity is the requirement.

In that case, TMOs (including Fe, Co, Ni, Cu, Mn, Zn, Mo, and Cr) offer a higher theoretical capacity, excellent cyclic performance, and higher operating voltages than graphite-based electrode materials [62,63]. The separation of metal lithium on the surface of TMOs is easier and possesses better safety. Their non-toxicity, high power density, natural abundance, and cost-effective fabrication process make them attractive anode material [64]. Moreover, using TMOs as anode material eliminates the problem of lithium dendrite formation, which is a major issue of the commercial graphite anode.

Among all TMOs, nickel oxide (NiO) is drawing attention and is widely explored as an alternative electrode for high-performance LIB because of its high theoretical capacity of 718 mAh g^−1^, cost-effectiveness, benign environmental effects, and natural abundance [65]. The density of NiO is 6.81 g cm^−3^, which is three times higher as compared with graphite of 2.26 g cm^−3^. The theoretical energy density of NiO is about 5.8 times higher than graphite anode [66]. During initial lithiation, a plateau voltage is evident at 0.6 V, whereas reversible voltage plateaus for discharging and charging are positioned at 1.3 and 2.2 V, respectively [67]. The corrosion resistance to alkalis, high melting point (1453 °C), ductility provided by the FCC crystalline structure, and adhesion to oxide coating generated on direct oxidation are some of the most notable structural features [68]. Moreover, NiO offers unique electrical, magnetic and optical properties.

However, NiO is a p-type, wideband gapped (~3.6 eV) semiconductor having a low electrical conductivity of less than 10^−13^ Ω^−1^ cm^−1^ and of distinct interest in certain applications such as solar cell or fuel cell composite anode [69], where specific mechanical properties are important to provide structural integrity to functional devices [70]. Nevertheless, the practical application of NiO in LIB is still obstructed because of its excessive volume variation and destruction of the NiO electrode structure as well as poor ionic conductivity, which results in poor electrochemical performance (poor cycle stability as well as rate capability) [71]. The conversion reaction of the NiO anode is given in the equation below:(3)$$\mathrm{NiO}\;+2{\mathrm{Li}}^{+}+2e^{-}\;↔\;\mathrm{Ni}\;+{\;\mathrm{Li}}_{2}O$$

Li_2_O is a potentially complicated species and can become electrochemically inactive if the size becomes too large [72], which leads to a drop in anode conductivity and capacity degradation. In addition, the excessive volume change results in the pulverization of active materials of NiO during the battery operation, which is the fundamental reason for the shortened life of LIB [73,74]. At the time of charging/discharging, the lithium insertion/de-insertion occurs in the active materials [75]. During insertion/de-insertion, the diffusion-induced stress and volume expansion/contraction causes micro-fracture [76] and results in capacity fading and an increase in internal impedance of LIB [77,78]. If the electrode materials are comparatively brittle, the effect of diffusion-induced stress is very significant. Since the material is brittle, cracks form within the material and it falls apart. This phenomenon eventually causes catastrophic failure of LIB, i.e., internal short-circuiting as two electrodes come in contact by puncturing the separator. The occurrence of internal short-circuiting causes 70% of the total cell energy to be released within 60 s [79] which increases the local temperature that stimulates the thermo-chemical reactions and leads to a runaway thermal effect [80]. These fundamental limitations can be overcome by reducing the particle size to the nano-level for a new kind of anode that is viable for the practical implementation of commercial LIB. Hence, this review also featured the electrochemical performance of Ni-NiO nanocomposite as an anode of LIB. Moreover, the benefits and disadvantages of employing nanocomposite are demonstrated as well.

## 4. Nanocomposite NiO as Anode of LIB

Downsides of graphite as the anode of LIB led to the search for a new anode which is suitable for increasing energy demand, cost-effective fabrication method, and environmental soundness and which can sustain structural integrity with a high specific capacity [81]. It is noted that the nanocomposite electrode is prepared through a slurry coating procedure whereby the electrode active material, binding agent, and conductive additives are amalgamated in a specific weight ratio to form a homogeneous mixture pasted on a copper foil (acting as a current collector). Traditionally, polyvinylidene di-fluoride (PVDF) is used as a binding agent, forming a hair-like structure that efficiently keeps the coating together. Carbon black has been used as conducting additive, improving electrode adhesion as well as reversible charge density, resulting in better processability [82]. There are several NiO based composite anode materials such as porous NiO [66], NiO nanowalls [67], NiO nanosheets [83], NiO nanogravel structure [84], NiO nanofibers [85], hollow nanotubes [86], macroporous NiO-films [87], mesoporous NiO-nanowires [88], mesoporous NiO crystals [89,90], hierarchical structures [91,92,93,94,95], and NiO/C composite [96,97,98,99] or conducting polymers [100,101] have been thoroughly investigated as LIB anode. These materials show promising results because of the nanostructural morphology, improved adhesion, and electronic conductivity.

Different morphological structures of NiO can be achieved through hydrothermal, microwave hydrothermal, co-precipitation, sol-gel method, solvothermal, chemical emersion, chemical vapor deposition, spray pyrolysis, and thermal oxidation of nickel powder. However, in this review, we will briefly discuss the nanocomposites of NiO with carbon, graphene nanosheet, carbon nanotube, reduced graphene oxide, and Ni as anode of LIB and the superiority of Ni-NiO nanocomposite over others.

### 4.1. Nanocomposite of NiO with Carbon

Pure NiO showed a specific capacity of 200 mAh g^−1^ at 0.5 C current rate after 40 charge/discharge cycles, whereas carbon-doped NiO-C composite exhibited an outstanding improvement in specific discharge capacity of 430 mAh g^−1^ for the same. This performance improvement of NiO-C composite was due to the carbon matrix which can absorb the volume expansion of NiO during cycling and to the porous microsphere architecture of NiO particles. Rahman et al. reported the synthesis of NiO-C nanocomposite, where carbon was surrounded by spherical shell clusters of NiO nanosized particles [102]. The fabrication procedure was via spray pyrolysis. The XRD pattern confirms the amorphous phase of carbon. The size of spherical NiO ranges from approx. 10–50 nm to greater than 500 nm. The galvanostatic charge/discharge test revealed that the nanocomposite NiO-C showed a higher capacity reservation which was 382 mAh g^−1^ for 50 cycles as compared to pure NiO which delivered 141 mAh g^−1^ for 50 cycles as shown in Figure 6, which was synthesized following the same method.

The enhanced capacity retention is attributed due to the NiO-C composite structure which is composed of carbon surrounded NiO nanoparticles, able to maintain the volume induced strain during charge/discharge and elevate the conductivity of NiO nanoparticles.

Huang et al. reported the synthesis of spherical-shaped NiO-C through hydrothermal dissemination of porous NiO within glucose solution and carbonization at 180 °C [103]. NiO-C composite showed 66.6% initial coulombic efficiency compared to 56.4% for NiO. In addition, NiO-C composite exhibited a higher specific capacity of 430 mAh g^−1^ after 40 cycles than NiO which had 200 mAh g^−1^ specific capacity. This improved electrochemical performance was due to the presence of the porous structure and conductive carbon. NiO-C composite not only can improve the specific surface area of elemental porous spheres but also maintain the electrical contact between the particles in spheres, which leads to improved electrochemical performance. A similar kind of nanostructured NiO/C was prepared via a one-pot hydrothermal synthesis method by Mu et al. [104]. The electrochemical performance of NiO/C nanocomposites exhibits a reversible capacity of 585.9 mAh g^−1^ after 50 cycles, which is much higher compared to the carbon-less NiO nanoparticles, as shown in Figure 7. Moreover, the nanoparticles of NiO/C showed significant discharge capacity, good rate capability, and remarkable cycle stability. The performance enhancement is due to the uniform coating of carbon on the NiO particles that improves the electrode conductivity and structural stability.

Guomin et al. prepared egg shell-yolk structured NiO/C porous composite via a two-pot hydrothermal method (Figure 8a) [105]. The result (Figure 8b–e) was that amorphous carbons existing in porous NiO/C, which offers a discharge capacity for the 100 cycles maintained at 625.3 mAh g^−1^; the capacity retention ratio is 94.1% relative to the 2nd discharge capacity.

The uniquely structured porous egg shell-yolk composite cushions the volume variation and restrains the agglomeration of active NiO material during cycling to improve material performance. The capacity becomes stable at 400.7 mAh g^−1^ with increasing current density to 800 mA g^−1^, displaying a good rate performance. The electrochemical performance improvement can be attributed to the porous and stable structure of the as-prepared composites.

Liu et al. synthesized an electrode of NiO/C nanocapsules where NiO nanoparticles and onion-like carbons are core and shell, respectively [106]. The NiO/C nanocapsule electrodes provide a discharge capacity of 1689.4 mAh g^−1^ at 0.5 C rate initially as well as sustaining a much higher reversible capacity of 1157.7 mAh g^−1^ after 50 cycles compared to NiO nanoparticles of 383.5 mAh g^−1^ specific capacity. The NiO/C nanocapsule as an electrode of rechargeable LIB shows an outstanding discharge capacity, a high rate capability, and exceptional cycling stability. The performance improvement can be attributed to the onion-like carbon shells. These are able to provide sufficient space to adapt to the volume change of NiO nanoparticles as well as to impede the solid electrolyte interface (SEI) film formation on the surface of the NiO nanoparticles.

### 4.2. Nanocomposite of NiO with Graphene Nanosheet

It is noted that when 3D-hierarchical NiO-graphene nanosheet (GNS) composite material is used as anode, the lithium storage ability is significantly improved with a high specific discharge capacity of 1400 mAh g^−1^. Figure 9a shows the enhanced cycle stability, and excellent rate capability. The composite can retain approximately 1065 mAh g^−1^ specific capacity, even after 50 cycles at 200 mA g^−1^ current density, as depicted in Figure 9b. The significant performance enhancement is because of the diminution of volume variation of NiO and improved electrical conductivity of the composite during cycling [107].

An investigation of hydrothermal synthesized NiO nanowalls/graphene nanosheets (NiO/GNSs) nanocomposites as anode materials of LIB was reported by Wang et al. [108]. The electrochemical result reveals that the NiO/GNS nanocomposites show very high capacity and cyclability, e.g., 844.9 mAh g^−1^ reversible capacity at 0.1C rate and little capacity fading (7.1%) after 50 cycles.

Shi et al. prepared a unique NiO@hollow carbon sphere (NiO@HCS) structure, which provides a huge electrode/electrolyte contact area and internal space for additional Li^+^ storage [109]. The thin-layered shells of porous carbon permit rapid diffusion of Li^+^ and electron movement. As an anode, the hollow structure provides an initial reversible capacity of 598 mAh g^−1^ at 0.1A g^−1^ current density and discharge capacity of 243 mAh g^−1^ after 400 cycles at a high current density of 2 A g^−1^. Further, Jae et al. synthesized raspberry-like hollow nanospheres of NiO, which were anchored on graphitic carbon sheets (GCS) [110]. The as-synthesized composite as an anode shows the enhanced electrochemical result with a higher reversible capacity of 1076.2 mAh g^−1^ at 0.1C after 50 cycles than that of bare NiO capacity, 9285.8 mAh g^−1^. In addition, the NiO/GCS composite is able to discharge high reversible capacities of 1073.6 mAh g^−1^ and 966.6 mAh g^−1^ after 300 cycles at 0.5C and 1C rates, respectively.

Fan et al. developed a facile and innovative approach to synthesize ultrathin NiO nanosheets@CMK-3 composite with nanosheets-mesoporous structure (Figure 10b) [111]. As an anode of LIB, the nanosheet composite (Figure 10a) shows a remarkable rate capability along with excellent cycling performance with an average specific capacity of 879 mAh g^−1^ from the second cycle and merely 9.8% capacity fading after 50 cycles at a 400 mA g^−1^ rate in comparison with bare CMK-3 and accumulated NiO nanosheets electrode (Figure 10c,d).

Chen et al. fabricated ultrafine nanocrystals of NiO, bonded with 3D graphene framework via in situ hydrothermal and subsequent annealing strategy [112]. The unique morphological structure shown in Figure 11 enables the hybrid electrode material to show an extremely high reversible capacity of 1104 mAh g^−1^ at 0.2 C rate after 250 cycles, and an excellent rate capability with 440 mAh g^−1^ specific capacity at 3 C rate as well as superior capacity retention during the charge/discharge process. Afterwards, a hydrothermal technique was adopted by Tian et al. to synthesize spherical porous Ni-MOFs material and convert it to NiO after subsequent calcination [113]. The reversible capacity of NiO material remains stable at 160 mAh g^−1^ at a 1C rate, and the coulombic efficiency reaches up to 97.12% at 200 cycles. At a constant current rate of 1C and 200 cycles, the reversible capacity reaches 440 mAh g^−1^, while the coulombic efficiency reaches 99.49%.

A chemically polymerized NiO-poly-pyrrole (NiO–PPy) composite was employed as anode for LIB, which exhibits enhanced electrochemical performance as compared to pristine NiO. The initial reversible capacities of the NiO and the NiO–PPy composite were 571 and 638 mAh g^−1^, respectively, which became 119 and 436 mAh g^−1^ after 30 cycles. The composite can retain 66% of capacity after 30 cycles. The result indicates that the electrochemical performance of the composite experienced remarkable enhancement (Figure 12). The cauliflower-like PPy is found to: enhance the conductivity of the prepared electrode; function as a binder improving the connection between particles; impede the aggregation of nickel region; buffer the volume induced strain during electrochemical cycling, and counteract the electrode cracking and pulverization [114].

### 4.3. Nanocomposite of NiO with Carbon Nanotube (CNT)

Carbon nanotubes are mainly 1-D cylindrical tubular graphite sheets possessing unique structure with high rigidity, high tensile strength, low density, and high conductivity. They can enhance capacity and be deployed as supporting matrix without any risk of pulverization [115,116].

Xu et al. reported a bamboo-like amorphous CNT-covered 1-D hierarchical NiO nanosheet composite which showed remarkable electrochemical performance as anode of LIB [117]. A high discharge capacity of 1034 mAh g^−1^ was delivered after 300 cycles at a relatively 800 mA g^−1^ current density and 98.1% coulombic efficiency. The specific reversible capacity improvement of the composite can be ascribed to the magnificent nanostructure, which results in synergistic effects of the amorphous hollow CNT and NiO nanosheets. In addition, a core-shelled NiO/carbon nanotube (CNT) microwire like composite as an alternative anode material for LIB showed a high reversible capacity of 752 mAh g^−1^ at 100 mA g^−1^ current density with 82% capacity retention after 30 cycles and superior rate performance with a specific capacity of 817 mAh g^−1^ at 100 mA g^−1^ after 5 cycles and 304 mAh g^−1^ at 1000 mA g^−1^ after 25 cycles as compared to pure NiO [118].

Bae et al. reported a self-assembled 3D graphene-carbon nanotube-nickel nanostructure [119], (Figure 13) which exhibited an initial capacity of 2395.2 mAh g^−1^. The 3D nanostructural composite delivered a high reversible capacity of 648.2 mAh g^−1^ after 50 cycles whereas the bamboo-shaped CNT electrode exhibited a reversible capacity of about 282.4 mAh g^−1^. The enhancement in performance within two electrodes may be attributed to the unique morphological characteristics of the 3D nanocomposite electrode. As the nanostructured composite possesses a high surface-to-volume ratio, this electrode showed high stability during lithiation/de-lithiation. This high stability arises from the mechanical flexibility of the graphene structure which causes reduction of the large volume variation.

The multiwalled carbon nanotube (MWCNT) NiO nanocomposite was prepared by using a thermal decomposition method [120]. The as-prepared composite samples were annealed at 250 °C/4 h, 300 °C/1 h, 350 °C/1 h, and 400 °C/1 h. The 300 °C/1 h sample exhibited a well-crystallized structure with initial discharge and charge capacities of 1083.8 and 720.2 mAh g^−1^, respectively, with 66.45% coulombic efficiency. This sample maintained a stable ~800 mAh g^−1^ discharge capacity and 97% coulombic efficiency after 50 cycles. However, for the sample at 250 °C/4 h, the capacity of discharge reached a stable value from 500–580 mAh g^−1^ before reaching 25 cycles and then exhibited a sharp decrease in the capacity in subsequent cycles. Moreover, for the sample annealed at 400 °C/1 h, the electrode delivered 1304.2 mAh g^−1^ initial discharge capacity corresponding to 3.64 Li/NiO sample and ~950 mAh g^−1^ stable discharge capacity from 2nd to 5th cycles. However, the loss of capacity increases sharply after the 5th cycle, with 50–100 mAh g^−1^ capacity loss/cycle as a result of electrode pulverization and electrical contact loss between the current collector and electrode active material. Consequently, the discharge capacity becomes 173.6 mAh g^−1^ at the 20th cycle and 13.3% of the capacity was retained as shown in Figure 14.

Zhang et al. synthesized NiO-graphene-carbon nanotubes (NiO-G-CNTs) by hydrothermal–thermal decomposition as shown in Figure 15 [121]. The nanohybrids delivered an initial discharge capacity of 1515.1 mAh g^−1^, a stable reversible specific capacity of 1022 mAh g^−1^ at 100 mA g^−1^ current density, and a specific capacity of 858.1 mAh g^−1^ after 50 cycles at 100 mA g^−1^ current density. The nanohybrid provides a 676 mAh g^−1^ specific capacity with an increased current density of 1000 mA g^−1^ and after 40 cycles. The improvement may be attributed to gradually activated graphene sheets in the nanohybrids during the cycling process.

A microwave-assisted chemical vapor deposition (CVD) method was employed for synthesizing biochar-CNT-NiO composite [122] as shown in Figure 16. The biochar-CNT-NiO exhibited an initial discharge capacity of 981.0 mAh g^−1^ with 65.18% coulombic efficiency. The presence of CNTs in the biochar-CNT-NiO led to the higher reversible capacity, which consequently inhibited the volume expansion of NiO, speeding the electron transport, and enhancing the performance of the battery. After 100 cycles, the biochar CNT-NiO exhibited 674.6 mAh g^−1^ capacity. Interestingly, the capacity of the composite was enhanced from the 30 to 100 cycles, due to gradual activation of both char and CNTs in the composite during the first several cycles. The composite electrode exhibits a rate capability of 522.7 mAh g^−1^ and 420.5 mAh g^−1^ at current densities of 1000 mA g^−1^ and 2000 mA g^−1^ as shown in Figure 16d.

### 4.4. Nanocomposite of NiO with Reduced Graphene Oxide

Reduced graphene oxide (RGO) incorporated with NiO nanoparticles can act as good matrix element for anode of LIB. It has some physical and chemical attributes in terms of electrical conductivity, thermal stability, surface area, and flexibility. Hence, the incorporation of RGO boosts up the electrical conductivity of active material and lessens the lithium ion transportation length. It also acts as support matrix that may inhibit the active materials of NiO droping off from the electrode and provide a cushion for extreme volume change during continuous electrochemical cycling.

Zhu et al. came up with RGO/NiO composite via homogeneous co-precipitation followed by subsequent annealing [97]. The prepared composite exhibited initial specific discharge/charge capacities of 1641 mAh g^−1^ and 1097 mAh g^−1^, respectively. Moreover, the composite exhibited tremendous cycling performance with a high specific discharge capacity of 1041 mAh g^−1^ after 50 cycles at 100 mA g^−1^ current density and an excellent rate capacity of 727 mAh g^−1^ at 1600 mA g^−1^ current density. The reduced graphene oxide and NiO composite show specific capacities of discharge and charge of 1641 and 1097 mAh g^−1^, respectively, which is higher as compared to the theoretical specific capacity of NiO. Furthermore, the composite exhibits a remarkable cycling performance, a high discharge capacity of 1041 mAh g^−1^ after 50 cycles at 100 mA g^−1^ current density, and excellent rate capacity with 727 mAh g^−1^ at 1600 mA g^−1^ current density [97].

A straightforward hydrothermal reaction combined with heat treatment was adopted to synthesize a nanocomposite of cross-linked rGO/NiO nanosheet [123]. This material as an anode of LIB exhibited outstanding results in terms of high discharge capacity, stable cyclic performance and remarkable rate capability. Three types of rGO/NiO nanocomposite were prepared by changing the content of NiCl_2_⋅6H_2_O. Among them, rGO/NiO-3 nanocomposite showed a specific discharge/charge capacity of 1570 and 1193 mAh g^−1^ with 75.6% coulombic efficiency (Figure 17). The coulombic efficiency value was maintained above 99.4% in the 3rd cycle. It shows remarkable rate capability with specific capacity of 756 mAh g^−1^ at 1.6 A g^−1^ current density. In addition, extraordinary cycle performance was achieved with 1141 mAh g^−1^ specific capacity, even after 130 cycles, which is 96.9% of the initial value. A stable capacity value of 1023 mAh g^−1^ was maintained over 200 cycles which showed the high cycle stability.

NiO was composited with rGO in a nanosheet-on-sheet-like architecture in a facile synthesis process [124]. As an anode of LIB this composite material exhibited high reversible capacity of 1036.8 mAh g^−1^ even after 50 cycles; because of the advantageous sheet on sheet nanostructure, the capacity presents a slower decrement during 50 cycles. The composite electrode shows remarkable lithium insertion/extraction capability with 97% coulombic efficiency from 4th cycles onward. Additionally, at high current density of 800 mA g^−1^, the nanocomposite is still able to deliver a specific capacity as high as 785 mAh g^−1^ and when the current density is turned back to 100 mA g^−1^, it can maintain a capacity close to the initial capacity. In addition, NiO was densely grown hydrothermally on the surface of rGO nanosheet with subsequent heat treatment in air [125]. The nanocomposite NiO/rGO (Figure 18) shows 1068 mAh g^−1^ specific capacity even after 100 cycles at a current density of 0.1 A g^−1^. Moreover, with increased current density of 2 A g^−1^, the electrode can deliver 870 mAh g^−1^ capacity, denoting a good rate capability of the nanocomposite electrode.

### 4.5. Nanocomposite of NiO with Ni

The aforementioned studies reveal that nanocomposites of NiO with carbon exhibit good cycle stability along with improved electrode conductivity due to the presence of carbon black/carbonaceous compound. However, its synthesis process is complicated and the low density of carbon significantly reduces the overall volumetric capacity. Therefore, the addition of high-density metals improves the electrochemical properties. The nanocomposite of NiO with metallic Ni nanoparticles can be a better option for improving the electrochemical characteristics.

The presence of metallic Ni facilitates the electrical conductivity that in turn causes an increase in the ion transfer. The catalytic activity of nanoparticle Ni also increases the reversible capacity as it accelerates the reverse reaction during charge through assisting the decomposition of the SEI layer as well as Li_2_O and formation of NiO that causes high coulombic efficiency and cycle stability. Du et al. demonstrated similar findings, revealing that the existence of elemental Ni favored electrical conductivity and offered preferable charge transfer kinetics [126]. Interestingly, only a few studies have been conducted on synthesizing nanocomposites which incorporate co-metal into the NiO matrix. Most of the research involved hybridizing with expensive carbon sources and tedious synthesis processes. Therefore, in this review we emphasized the Ni-NiO nanocomposites and tried to ascertain their electrochemical performance. Kozlovskiy et al. [127] reported an ionizing radiation method to enhance the efficiency of performance of Ni-NiO nanotubes as anodes of LIBs. The Ni-NiO nanotube structured anode exhibited a stable discharge capacity of 900 mAh^−1^ after three charge/discharge cycles. Furthermore, Kozlovskiy et al. [128] reported an increase in Ni nanotubes’ lifetime when used as anode of LIBs. It is noted that modified nanostructures exhibited an increase in the number of cycles from 344 to 607 and 651. The increase in the lifetime of the anode material for irradiated samples was due to the removal of stresses and distortions in the crystal lattice, as well as partial annealing of defects as a result of irradiation [129,130,131].

Meng et al. reported a facile approach to fabricate a unique 3D NiO-Ni nanowire composite (Figure 19a–c) attached to the carbon deposited eggshell membrane [132]. The membrane performed various tasks such as supplying active sites for nucleation and development of nanowires of Ni(OH)_2_ and delivered a reducing medium for partial reduction of NiO into Ni. The as-prepared 3D composite nanowires on carbon deposited eggshell membrane showed outstanding rate performance and cycling ability in comparison to bare nanowires of NiO (Figure 19e). The electrodes offered a capacity of 827 mAh g^−1^ at the 10th cycle, at 100 mA g^−1^ current density. The specific capacity reaches 424 mAh g^−1^ at the 40th cycle, at 1000 mA g^−1^ current density, and can retain stable capacity up to 900 mAh g^−1^. On the other hand, NiO showed poor cycling and rate performances. The superior result enhancement of the composite anode could be attributed to the existence of Ni nanoparticles (Figure 19d) as well as carbon deposited eggshell membrane.

The formation of in-situ Ni nanoparticles on the wires of NiO can improve the lithiation/de-lithiation mechanisms and SEI formation during cycling. According to the studies, these 3D architectures of nanowires with eggshell membranes, are able to sustain volume variations and improve electrode structure and stability. Figure 19c shows that the nanowire electrode system was well preserved on the surface of the eggshell membrane after subsequent cycles. In addition, Liang et al. reported Ni-NiO nanocomposite encapsulated on the surface of carbon block via hydrothermal methods [133]. The increase in specific capacity of the composites was due to high catalytic activity and the synergistic effect of transition metals and their oxides. The nanocomposite electrode exhibited a reversible capacity of 1200 mAh g^−1^ at 2C current density even after 650 cycles which demonstrates excellent cycle stability.

Zhang et al. prepared Ni@NiO via low-temperature pyrolysis, and lithium storage performances were analyzed [134]. The nanocomposite delivered a 1st cycle high specific charge capacity of 827.7 mAh g^−1^ with 77.1% coulombic efficiency at 0.1 A g^−1^ current density when 0–3 V voltage was supplied. Moreover, a facile pyrolysis technique was adopted to prepare Ni-NiO nanocomposite dispersed on carbon matrix, and a stable lithium storage performance was evaluated [135]. The nanocomposite electrode delivered 400 mAh g^−1^ discharge capacity at 0.1C rate and a stable capacity of 100 mAh g^−1^ at 0.3C rate was maintained for 500 cycles, with 70% discharge capacity retention. Suhang et al. prepared a unique scale-like Ni-NiO nanocomposite following a solvothermal method [136]. The nanocomposite revealed high specific discharge and charge capacity of 1224.4 and 782.7 mAh g^−1^ at 200 mA g^−1^ current density, superior capacity retention of 107.92% even after 80 cycles and outstanding cycle stability with 552.8 mAh g^−1^ at 500 mA g^−1^ current density after 200 cycles.

Yue et al. introduced a 3D-flower-like nanocomposite (NiO, NiO@C, and NiO/Ni) through the calcination process [137]. This nanocomposite (Figure 20) delivered an initial discharge and charge capacity of 1071 mAh g^−1^ and 894 mAh g^−1^, respectively, which is better than bare NiO. However, the carbon-coated 3DNiO@C and Ni-doped NiO/Ni nanocomposite exhibited discharge/charge of 985 mAh g^−1^ and 805 mAh g^−1^; 1316 mAh g^−1^ and 898 mAh g^−1^, respectively. Moreover, the NiO/Ni and NiO@C showed stable reversible capacity, better capacity retention, superior cyclability, and rate performance than the nanoflower-like NiO. Since the C and Ni themselves possess good electronic conductivity, their presence keeps a good contact between NiO particles and the current collector, facilitating the ion/charge transport.

Huang et al. prepared NiO-Ni nanocomposite by a simple method and the electrochemical characteristics of the nanocomposite as an anode for LIB were evaluated [138]. The XRD diffraction pattern showed the sample prepared in the air was a combination of Ni and NiO; the sample prepared in oxygen was fully NiO. The as-prepared NiO-Ni composite exhibits a first discharge capacity of 1152.4 mAh g^−1^ that is higher compared to the NiO electrode 997.4 mAh g^−1^, as shown in Figure 21. The initial coulombic efficiency of the composite anode (71.2%) is higher than the traditional NiO (64.9%). The nanocomposite shows much higher reversible capacities and cycling ability than the bare NiO.

The improved results could be due to the presence of nanoscale Ni in the composite. In the case of NiO-Ni composite, the decomposition of amorphous Li_2_O, as well as SEI with poor electronic conductivity, boosts the conductance of material and the metallic Ni phase increases the electrical conductivity. Additionally, the defects in the nanocomposite structure provide sites for Li^+^ insertion and facilitate ion diffusion.

Xia et al. prepared spherical NiO/Ni nanocomposite via a facile citric–nitrate method with in situ generation of Ni [139]. The TEM image in Figure 22a,b reveals that the NiO/Ni nanocomposite has 30–40 nm-sized uniformly distributed particles.

The as-prepared nanocomposite as an anode exhibits improved electrochemical performance with high reversible capacity, excellent rate capability, and significantly long cycle stability, with a high specific capacity of 800 mAh g^−1^ after 50 cycles, at a current density of 0.1C (Figure 22d). Moreover, the NiO/Ni electrode shows a reversible capacity of 450 mAh g^−1^, even at a high current density of 5C and a capacity of 635 mAh g^−1^ for 300 cycles at 2C, at a temperature of 50 °C (Figure 22e). The research outcomes of the discussed NiO-based composites are summarized in Table 2.

The Ni-NiO nanocomposite has been studied thoroughly as an anode due to its good cyclic ability, improved rate capability, and high capacity retention ability. Furthermore, it delivered high reversible capacity and enhanced coulombic efficiency compared to other anode materials. However, NiO undergoes poor cyclability and low electrical conductivity issues due to the semiconducting nature of NiO particles and volume-induced strain during the electrochemical process. The concept of doping Ni into NiO enhanced the electrochemical performance as the excess nanosized metallic Ni particles provide a highly conductive medium and catalytic activity facilitating the better decomposition of Li_2_O during charge. The volume-induced problems can be resolved through the implication of different nanostructural NiO doped with Ni.

The binder-involved nanocomposite has some issues such as detachment of active particles from the current collector caused due to weak bonding between conventionally used PVDF binder and active NiO particles [96,102]. PVDF binder is used most frequently for anode/cathode preparation due to its inherent properties such as superb thermal and electrochemical stability, and strong bonding between electrode particles and current collector foil [141,142]. However, its practical implication is obstructed because of its poor flexibility, being easily inflamed with temperature rise, and requiring use of an organic solvent—most commonly N methyl-2-pyrrolidone (NMP), which is costly, toxic, has poor flexibility and is inflammable. Therefore, replacing PVDF with other binders such as styrene-butadiene rubber (SBR) and carboxy methyl cellulose (CMC) offers significant improvement in interfacial bonding, cycle stability, capacity retention ability, and rate capacity [143]. However, the slurry-coated nanocomposite synthesis route is very facile and simple whereas the synthesis of binder-free NiO foam nanocomposite electrode requires an argon environment, while the pore size and thickness of the foam also need to be controlled [144]. In addition, the binder-free material suffers from poor electronic conductivity problems [145] whereas the binder incorporates conductive carbon and parental metal powder which facilitates the electronic conduction of the electrode material.

## 5. Conclusions

This review focused on the binder-employed NiO nanocomposite as an anode of high-performance LIB. Extensive research has been carried out on bindered nanocomposite of NiO incorporating binder material such as PVDF. This improves the adhesive and cohesive bonding between electrode active particles or electrode particles and current collector, and the conducting agent (Super P-carbon black) enhances the conductivity of the electrode material. The reasons for replacing the conventional graphite anode and choosing TMO anode-NiO has been explained. In this review, we have epitomized different types of nanocomposite NiO and some critical points regarding this study are pointed out below:As compared to commercial graphite, NiO has two times higher specific capacity, is environmentally friendly, and has high stability. When NiO is incorporated with carbon to synthesize nanocomposite it shows enhanced cycling performance. In addition, NiO with CNT, MWCNT, RGO, GNS, HCS, etc. exhibited remarkable improvements in performance because these carbon structures provide cushioning and act as a support matrix, hence improving the conductivity as well as impeding the detachment of active material from the electrode.The doping of Ni into NiO causes the formation of Ni-NiO nanocomposite which facilitates the ionic conductivity of the material due to the presence of metallic Ni within the structure. The presence of metallic nickel assists the reverse reaction during charging decomposition of SEI and Li_2_O. The incorporation of Ni into NiO shows better electrochemical performance than the carbonaceous compound.When Ni-NiO nanocomposite powder is used as anode the powder structure can smoothly buffer the volume change as a result of continuous expansion/contraction. Therefore, performance enhancement and increased conductivity can be observed. The electrode surface area, electrode/electrolyte contact, and short Li^+^ diffusion distance can be increased due to the nanocomposite structure.

The binder-free anode confronted issues such as low thermal conductivity and poor cycle performance for the semi-conductive nature of NiO. The binder-free anode i.e., NiO foam nanocomposite fabrication involved precise chemical procedures, controlling different parameters; an argon environment is also essential to fabricate NiO foam from powdered Ni, which is very tedious and expensive. In contrast, the slurry-coated conventional procedure is simple, facile, and comprises parent metal, conductive material which significantly increases the electronic conduction. The issues faced in bindered anode can be sorted out through the replacement of conventional binder with a possible high-quality binding material.

While the inclusion of the Ni phase improves the electrochemical properties of the nanocomposite, extensive research is still needed for further improvement in the electrode material of LIB. In particular, further research is required in the field of (i) the formation mechanism of NiO on the metal powder, (ii) the storage and kinetic transportation mechanism of the nanocomposite materials, (iii) the improvement of bonding between active particles and current collector, (iv) the effects of varying the weight ratio of the slurry constituents, and (v) establishing a facile and scalable synthesis route.

## Figures and Tables

**Figure 1 nanomaterials-12-02930-f001:**
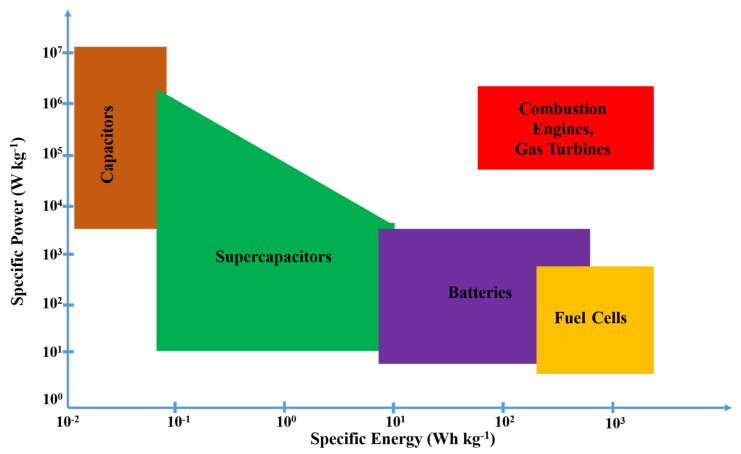
Specific energy and specific power plot of different energy storage systems. Reproduced with permission from [5].

**Figure 2 nanomaterials-12-02930-f002:**
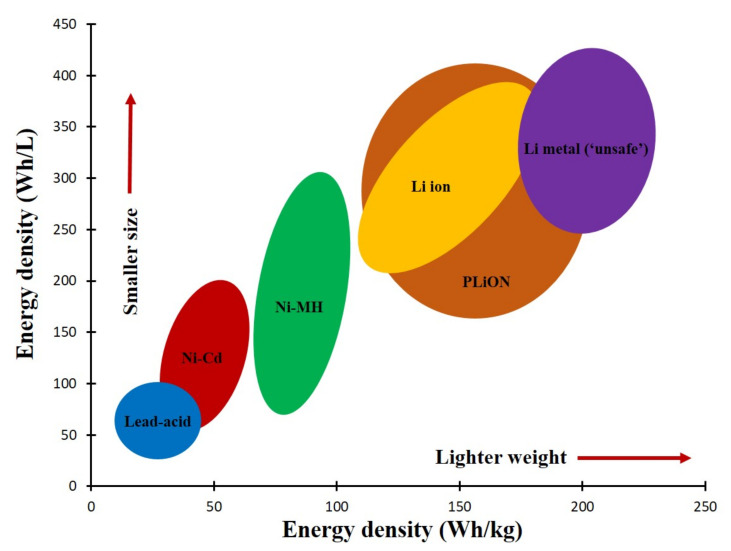
Comparison between LIBs and other batteries in terms of energy densities. Reproduced with permission from [9].

**Figure 3 nanomaterials-12-02930-f003:**
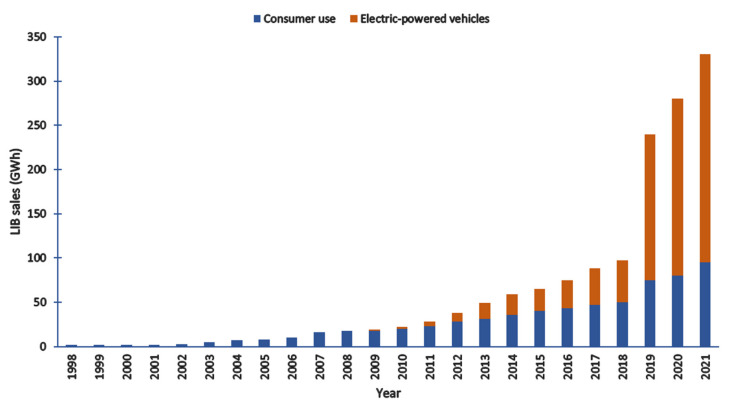
Demand for LIB for consumer use and electric vehicles in two decades. Reproduced with permission from [16].

**Figure 4 nanomaterials-12-02930-f004:**
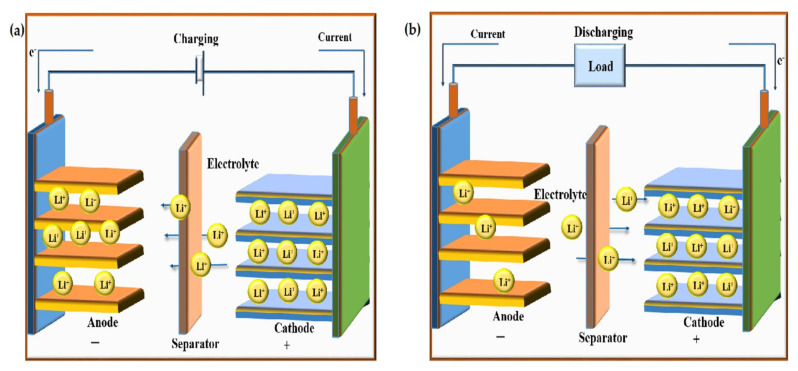
Illustration of the operating principle. (**a**) Charging and (**b**) discharging of a typical Li-ion battery cell.

**Figure 5 nanomaterials-12-02930-f005:**
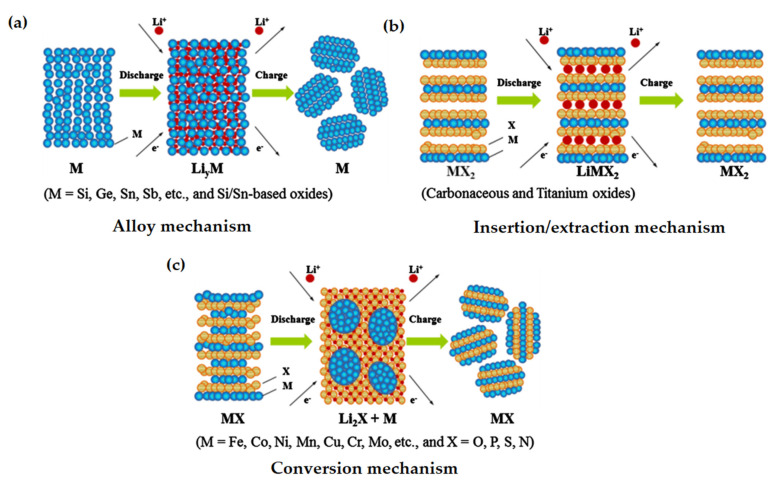
Reaction mechanism of metal oxides. (**a**) Alloy, (**b**) Insertion/extraction and (**c**) Conversion. Adapted with permission from [36].

**Figure 6 nanomaterials-12-02930-f006:**
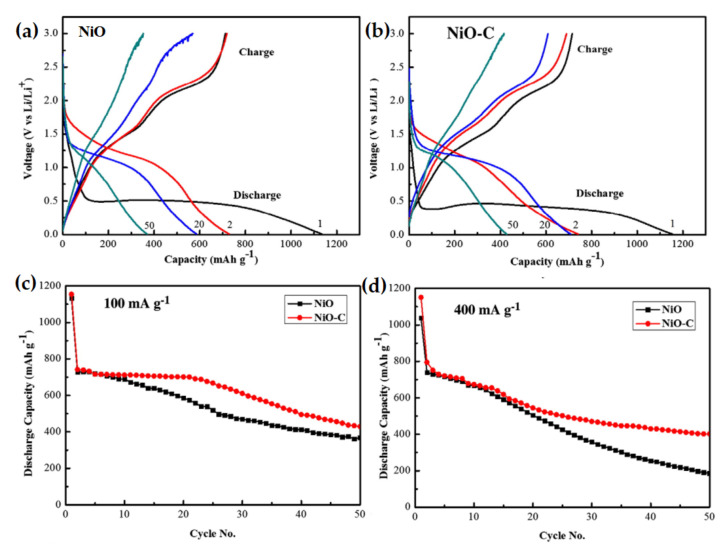
Galvanostatic discharge/charge profiles of (**a**) NiO and (**b**) NiO-C at 100 mA g^−1^. Capacity retention characteristics of NiO and NiO-C anode with cycle no. at different current densities of (**c**) 100 mA g^−1^, and (**d**) 400 mA g^−1^. Adapted with permission from [102].

**Figure 7 nanomaterials-12-02930-f007:**
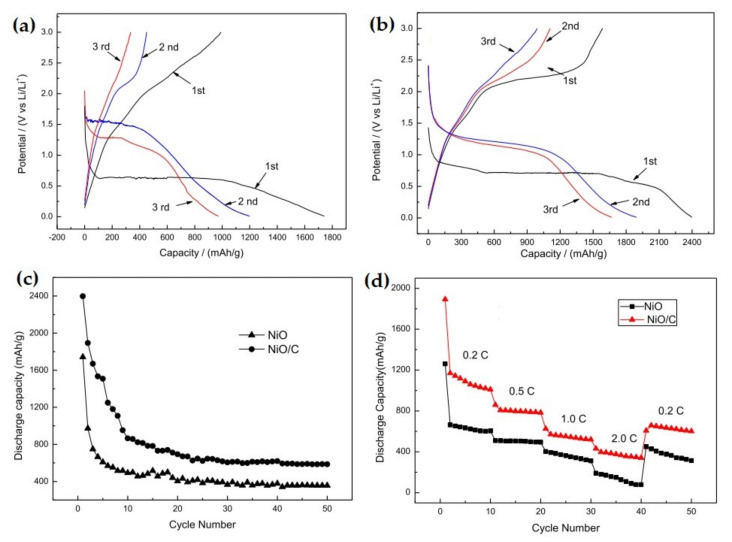
Charge/discharge curves of (**a**) NiO, and (**b**) NiO/C composite at 70 mA g^−1^; (**c**) Cycle stability of NiO and NiO/C composite at 70 mA g^−1^; (**d**) Rate capability of NiO and NiO/C composite. Adapted with permission from [104].

**Figure 8 nanomaterials-12-02930-f008:**
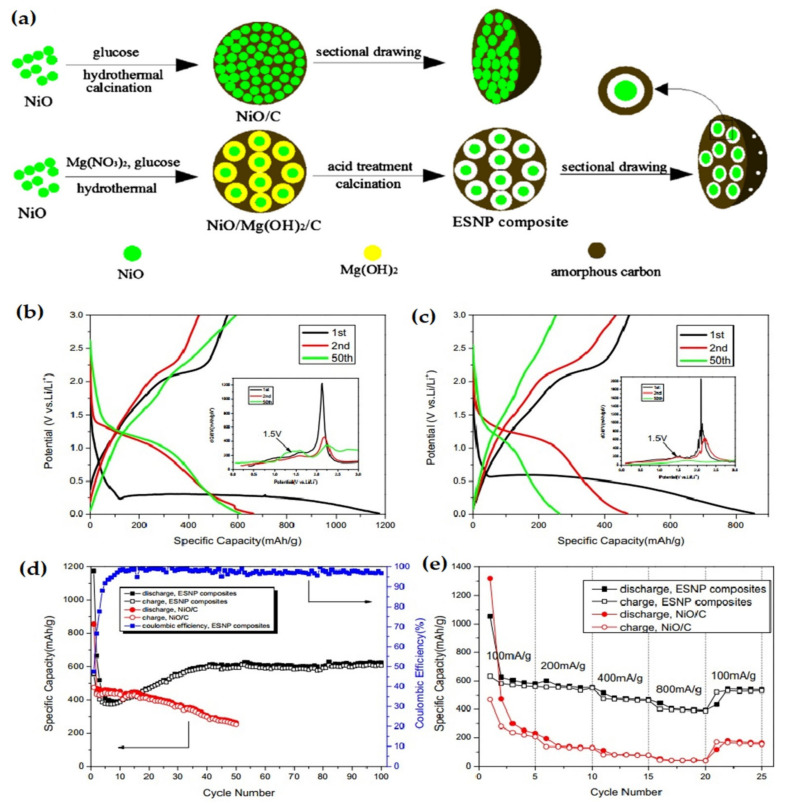
(**a**) Schematic view of synthesis of egg-yolk shell NiO/C porous composites. Electrochemical performance of (**b**,**d**) egg-shell yolk structure porous NiO/C composite. (**c**,**e**) NiO/C. Adapted with permission from [105].

**Figure 9 nanomaterials-12-02930-f009:**
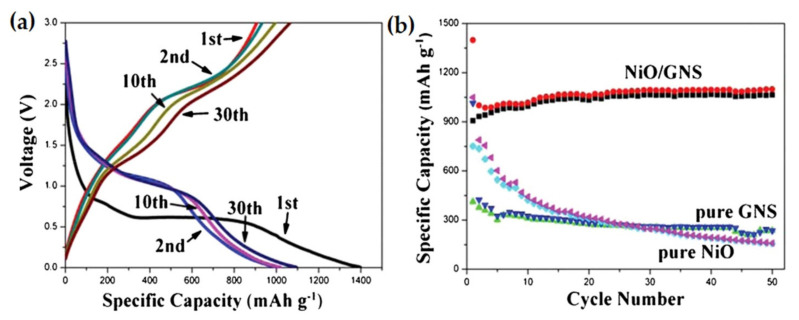
The charge-discharge profile of (**a**) 3D-hierarchical NiO-GNS composites. (**b**) The comparison of the cycling performance of composites, GNS, and NiO. Adapted with permission from [107].

**Figure 10 nanomaterials-12-02930-f010:**
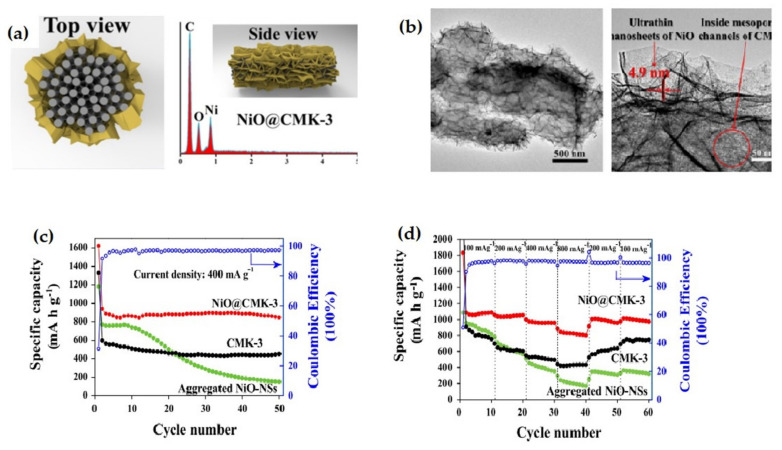
(**a**) Top and side view of NiO@CMK-3 composites (inset EDX data). (**b**) TEM images of the composites showing nanosheets of NiO. (**c**) Cycle performance at 400 mA g^−1^ current density. (**d**) Rate capability at distinct current density. Adapted with permission from [97].

**Figure 11 nanomaterials-12-02930-f011:**
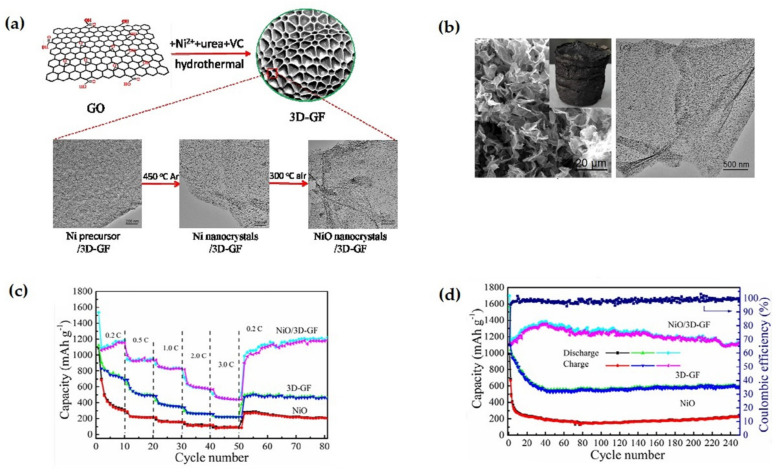
(**a**) Schematic view of the fabrication process of NiO/3DGF. (**b**) SEM and TEM images of nanohybrids. (**c**) Rate capability of the nanocomposite three electrodes at distinct current rates. (**d**) Cycling performance comparison and coulombic efficiency of nanocomposite. Adapted with permission from [112].

**Figure 12 nanomaterials-12-02930-f012:**
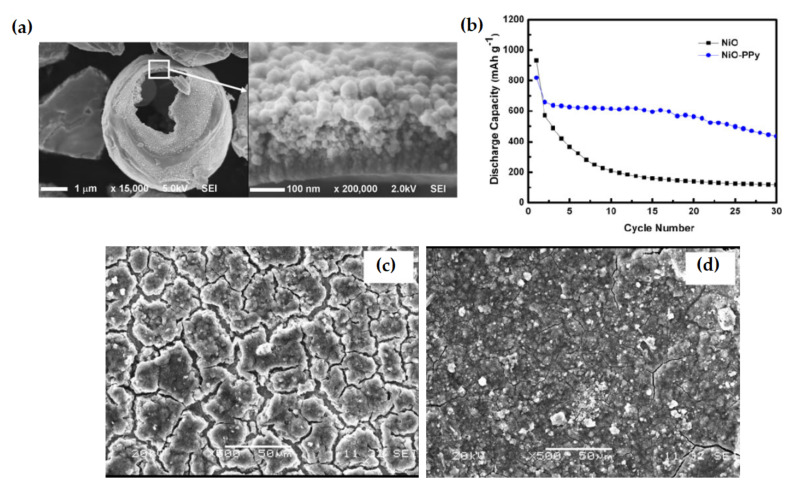
(**a**) FESEM image of NiO–PPy composite. (**b**) Discharge capacity of NiO and NiO–PPy electrodes corresponding to cycle no. (**c**,**d**) SEM images of NiO and NiO–PPy electrodes after 30 cycles. Adapted with permission from [114].

**Figure 13 nanomaterials-12-02930-f013:**
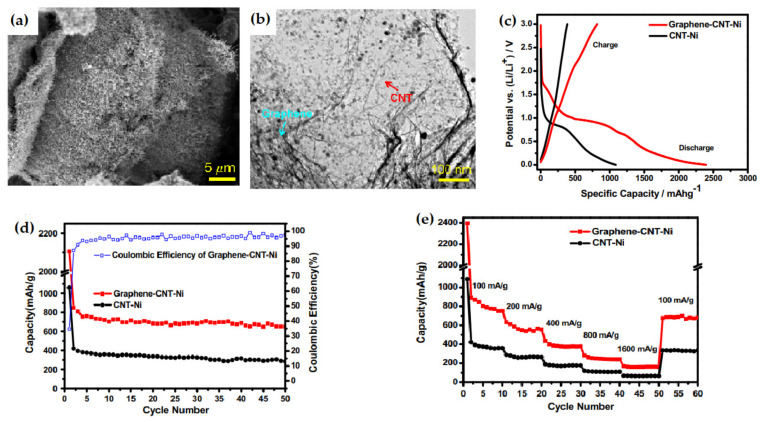
(**a**) Image showing densely grown CNTs over graphene sheets. (**b**) Image showing curled CNTs on wrinkled paper-like graphene sheets. (**c**) Galvanostatic charge/discharge profiles of nanostructures and bamboo-shaped CNTs at 100 mA g^−1^ current density. (**d**) Cycle performance of 3D G-CNT-Ni nanostructures and bamboo-shaped CNTs at 100 mA g^−1^ current density. (**e**) Rate capability of 3D G-CNT-Ni nanostructures and bamboo-shaped CNTs. Adapted with permission from [119].

**Figure 14 nanomaterials-12-02930-f014:**
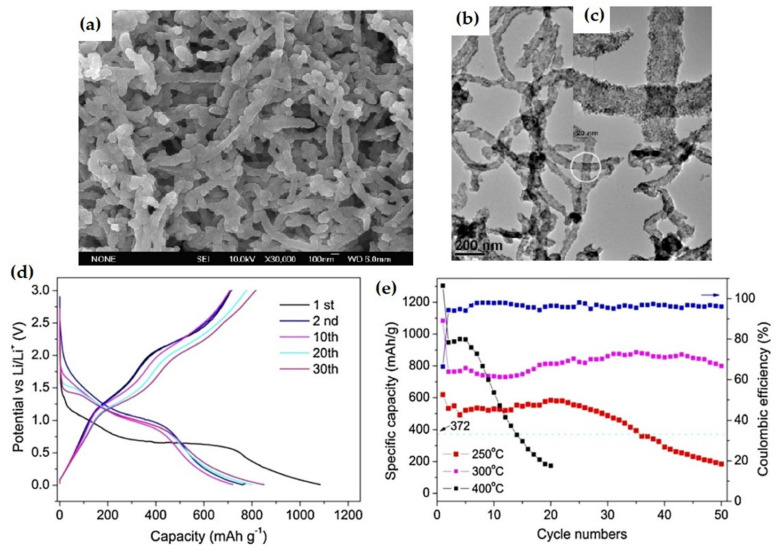
(**a**) SEM image of the NiO/MWCNT composites annealed at 300 °C/1 h. (**b**,**c**) TEM images of the composite (300 °C/1 h). (**d**) Galvanostatic discharge/charge profiles of NiO/MWCNT composite. (**e**) Cycle performance of the composite structure. Adapted with permission from [120].

**Figure 15 nanomaterials-12-02930-f015:**
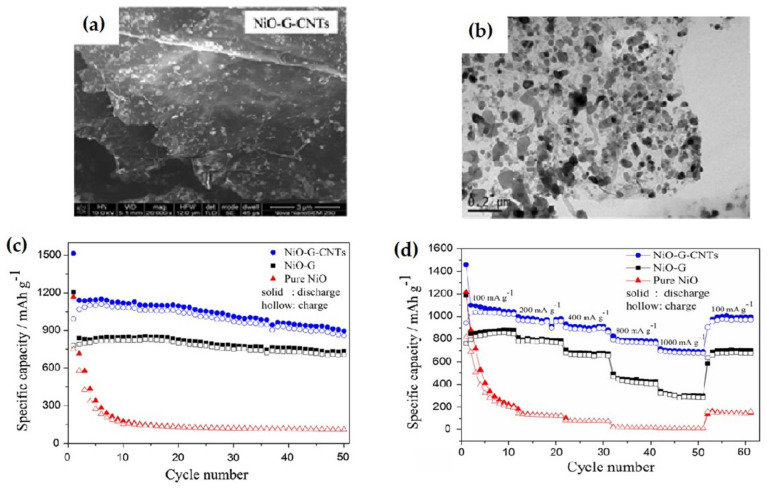
(**a**,**b**) SEM and TEM images of 3D NiO–G–CNTs. (**c**) Cycling performances of NiO–G–CNTs, NiO–G, and pure NiO electrodes at specific current density of 100 mA g^−1^. (**d**) Rate capability of the NiO–G–CNTs, NiO–G, and NiO electrodes at distinct current densities. Adapted with permission from [121].

**Figure 16 nanomaterials-12-02930-f016:**
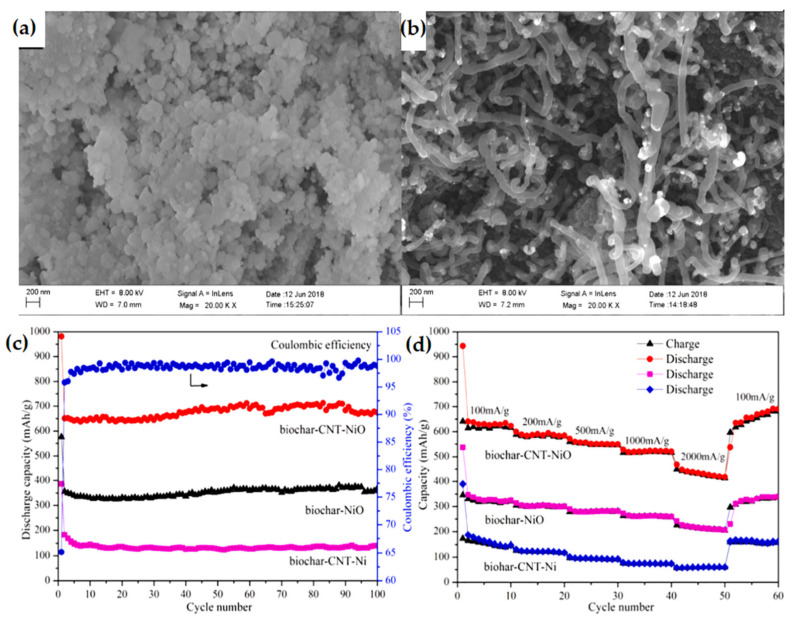
(**a**,**b**) SEM images of biochar-CNT-NiO. (**c**) Cycle performance of biochar-CNT-NiO, biochar-NiO, and biochar-CNT-Ni at specific current density of 100 mA g^−1^. (**d**) Rate performance at five distinct current densities. Adapted with permission from [122].

**Figure 17 nanomaterials-12-02930-f017:**
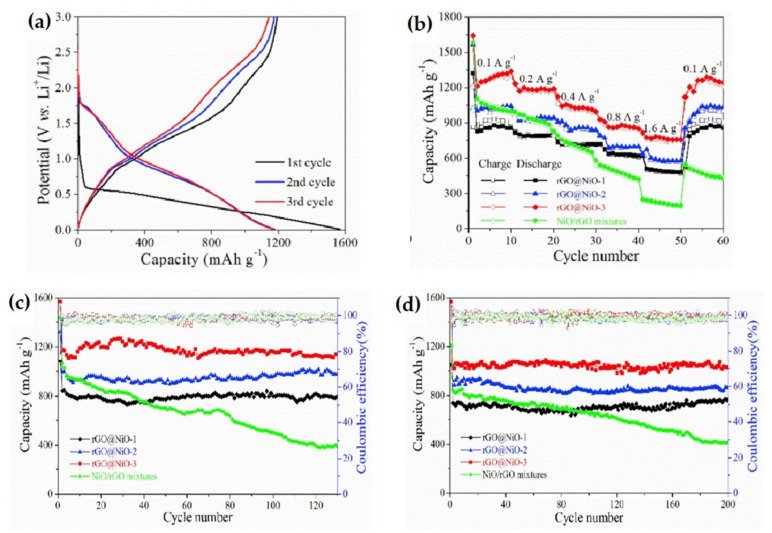
(**a**) Galvanostatic charge/discharge profile of rGO/NiO-3 nanocomposite. (**b**) Rate performance at five different current densities. (**c**,**d**) Cyclic performance of nanocomposite at 100 and 400 mA g^−1^ current density. Adapted with permission from [123].

**Figure 18 nanomaterials-12-02930-f018:**
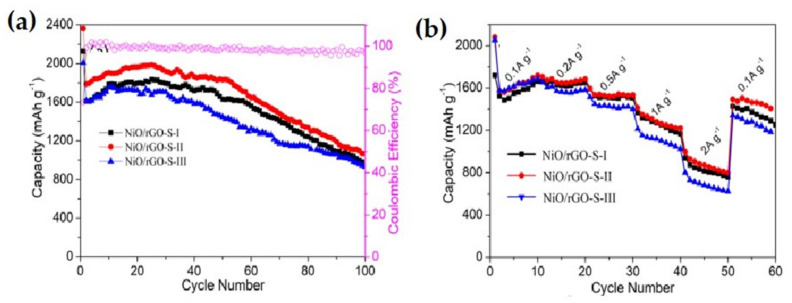
(**a**) Cycling performance and columbic efficiency of nanocomposite NiO/rGO containing different quantities of NiO. (**b**) Rate capability the NiO/rGO composite. Adapted with permission from [125].

**Figure 19 nanomaterials-12-02930-f019:**
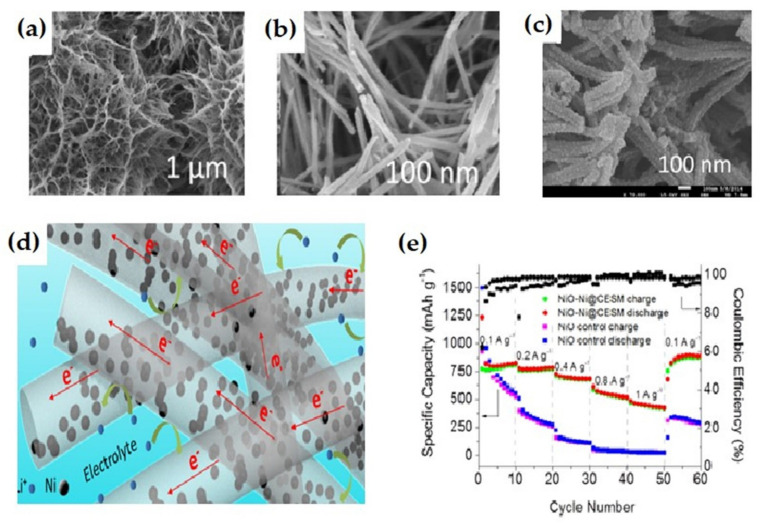
FESEM image of a-Ni(OH)_2_ nanowires on an egg-shell membrane at (**a**) low magnification, (**b**) high magnification. (**c**) Morphology of nanowire after cycling (well preserved). (**d**) Illustrative of role conducted by nano-sized metallic Ni and 3D network in promoting lithium storage. (**e**) Rate performance of NiO-Ni@CESM and NiO at various C-rates. Adapted with permission from [132].

**Figure 20 nanomaterials-12-02930-f020:**
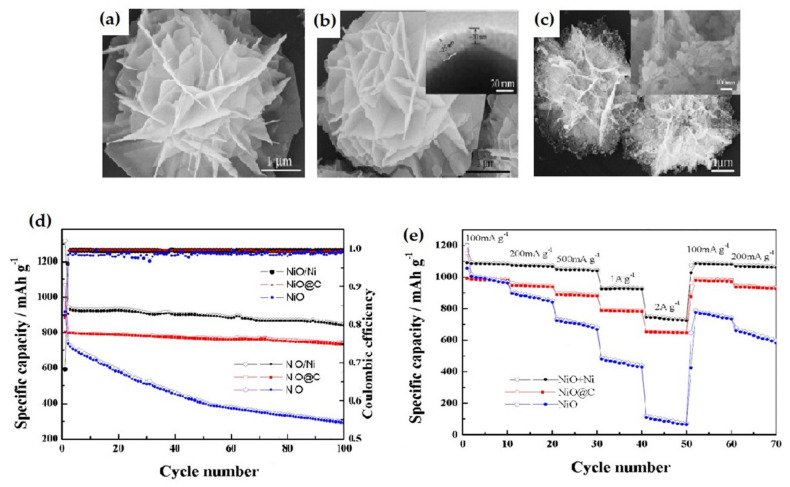
SEM images of (**a**) pure NiO, (**b**) NiO@C nanocomposite, (**c**) NiO/Ni nanocomposite. (**d**) Cycling performances of the bare NiO, NiO@C, and NiO/Ni nanocomposites at a current density of 1 A g^−1^. (**e**) Rate performances of the bare NiO, NiO@C, and NiO/Ni nanocomposite. Adapted with permission from [137].

**Figure 21 nanomaterials-12-02930-f021:**
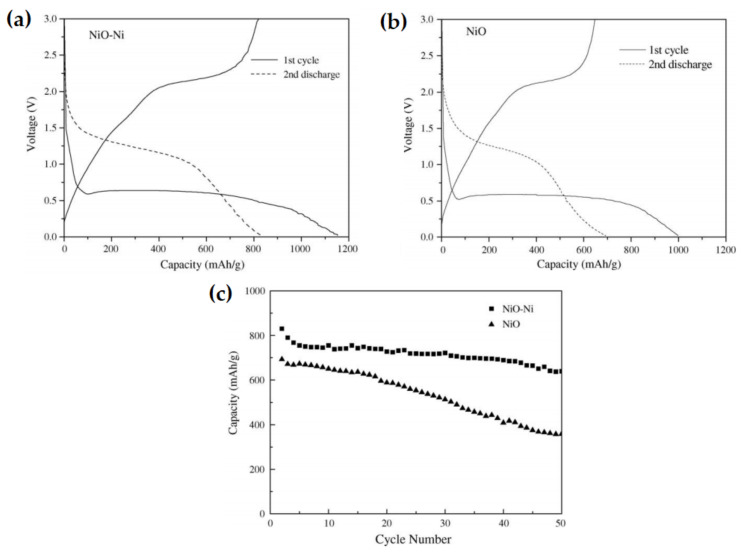
Curves of the first cycles and the second discharge for (**a**) NiO–Ni nanocomposite and (**b**) NiO. (**c**) Cycling performances for the NiO–Ni nanocomposite and NiO (2–50th cycle). Adapted with permission from [138].

**Figure 22 nanomaterials-12-02930-f022:**
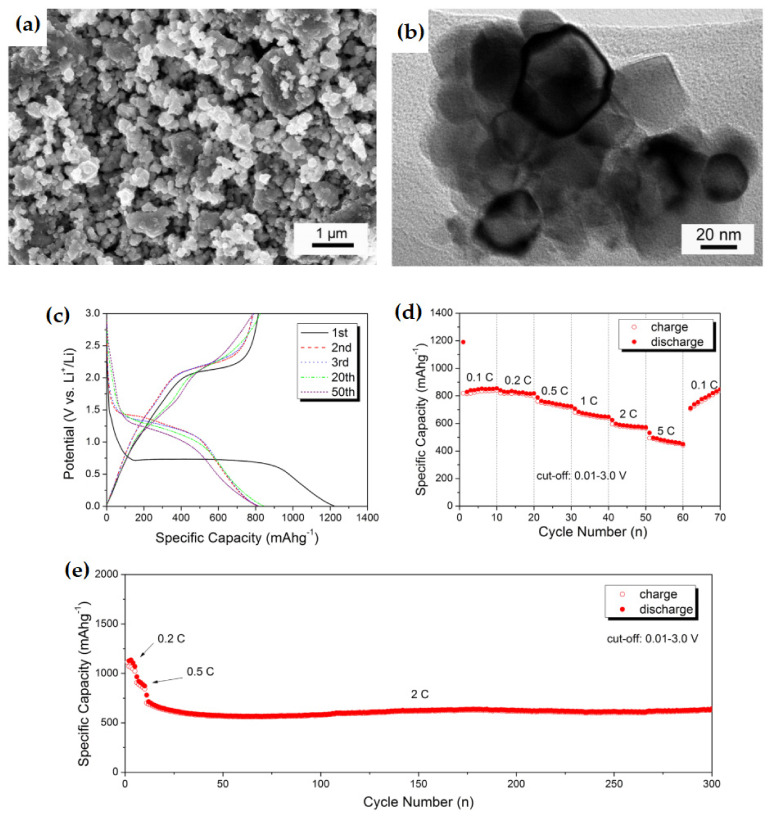
TEM (**a**) low and (**b**) high magnification images of NiO/Ni nanocomposite anode. (**c**) Charge-discharge profile of certain cycles. (**d**) Rate performance and capacity retention ability. (**e**) Cycling performance at 50 °C and 2C rate. Adapted with permission from [139].

**Table 1 nanomaterials-12-02930-t001:** Electrochemical Performance of some Silicon nanocomposite electrode.

Materials	Specific Capacity (mAh g^−1^)	Remarks	Ref.
Si/graphite composite with polymer microsphere	Charge and discharge capacity of 1493 mAh g^−1^ and 1091 mAh g^−1^, respectively with 73.0% 1st cycle coulombic efficiency.	The capacity drops rapidly with continuous cycling and became approx. 790 mAh g^−1^ at 50 cycles.	[46]
Si/porous-C composite with voids	A reversible capacity of 980 mAh g^−1^ after 80 cycles, little capacity decay per cycle (0.17%), excellent rate capability of 721 mAh g^−1^ at a high current density of 2000 mA g^−1^.	Rapid capacity depletion as a result of crack formation and mechanical degradation of active electrode material over cycling.	[47]
3D-Carbon fiber/Si nanocomposite	The reversible capacity in the 1st cycle at a 0.05 C rate was between 2.5 Ah g^−1^ and 3 Ah g^−1^.	High reversible specific capacity, low cyclability.	[48]
Raspberry-like HSi/C nanocomposite	Reversible specific capacity of 886.2 mAh g^−1^ at 0.5 A g^−1^ current density after 200 cycles with high rate capability and cycle ability of 516.7 mAh g^−1^ at 2 A g^−1^ after 500 cycles.	The specific capacity decreases rapidly with cycling and increasing current density and electrodes become pulverized.	[49]
Si/graphene nanocomposite	The initial discharge capacity of 769 mAh g^−1^, at 4000 mA g^−1^ current rate.	Improved reversible specific capacity, low initial capacity, capacity decreased with high C-rate.	[50]
Dual yolk-shell Si/C structure	Stable specific capacity of 956 mAh g^−1^ at 0.46 A g^−1^ after 430 cycles with 83% capacity retention.	Reversible capacity, capacity drastically decays with cycling.	[51]

**Table 2 nanomaterials-12-02930-t002:** Summary of the electrochemical performance of NiO-based nanocomposite anodes for high performance LIB.

Materials	Specific Capacity (mAh g^−1^)	Remarks	Ref.
NiO-C nanocomposite	A high initial capacity of 1102 mAh g^−1^. After 50 cycles, 37% of initial discharge capacity was retained.	High initial specific capacity, small capacity retention ability.	[102]
Net-structured NiO-C	A reversible capacity of 429 mAh g^−1^ even after 40 cycle at 71.8 mA g^−1^ current density.	Stable cycle performance due to carbon inclusion.	[140]
Spherical shaped NiO-C nanocomposite	High specific capacity of 430 mAh g^−1^ after 40 cycles with 66.6% initial coulombic efficiency at 0.5C rate.	Good cyclic performance and high initial coulombic efficiency.	[103]
NiO/C nanocomposite	A high reversible capacity of 585.9 mAh g^−1^ after 50 cycles.	High specific discharge, remarkable cyclic stability and good rate performance.	[104]
Egg shell-yolk structured NiO/C porous composite	The first specific discharge capacity was 1175.2 mAh g^−1^ with 0.22 V discharge voltage. It maintained 625.3 mAh g^−1^ capacity after 100 cycles.	High capacity retention ability, good rate capability.	[105]
NiO/C nanocapsules	Initial discharge capacity of 1689.4 mAh g^−1^ at 0.5 C rate with a high reversible capacity of 1157.7 mAh g^−1^ after 50 cycles.	Outstanding discharge capacity, high rate capability, and exceptional cycling stability	[106]
3D-hierarchical NiO-graphene nanosheet (GNS) composite	A high specific discharge capacity of 1400 mAh g^−1^. Even after 50 cycles, the composite can retain 1065 mAh g^−1^ specific capacity at 200 mA g^−1^ current density.	High discharge capacity, outstanding rate performance.	[107]
NiO nanowalls/GNS nanocomposites	A high reversible capacity of 844.9 mAh g^−1^ at 0.1C rate with little capacity fading of 7.1% after 50 cycles.	High capacity, cyclic stability and little capacity decay.	[108]
NiO@hollow carbon sphere	The structure provides an initial reversible capacity of 598 mAh g^−1^ at 0.1A g^−1^ current density. Even after 400 cycles delivers discharge capacity of 243 mAh g^−1^ at high current density of 2 A g^−1^.	Outstanding reversible capacity, stable cycle performance and rate capability.	[109]
Hollow nanospheric NiO/GCS	High reversible capacities of 1073.6 mAh g^−1^ and 966.6 mAh g^−1^ after 300 cycles at 0.5C and 1C rates	Highly reversible capacity, excellent cyclic performance and rate capability.	[110]
NiO nanosheets@CMK-3 composite	The composite delivers discharge and charge capacity of 1641 and 1097 mAh g^−1^, merely 9.8% capacity fading after 50 cycles at a 400 mA g^−1^ rate.	High average specific capacity, remarkable cycling performance, excellent rate capacity.	[111]
NiO/3DGF nanocomposite	It shows an extremely high reversible capacity of 1104 mAh g^−1^ at 0.2C rate after 250 cycles, and an excellent rate capability with 440 mAh g^−1^ specific capacity at 3C rate.	Highly reversible capacity, excellent rate performance, superior capacity retention.	[112]
NiO–PPy composite	The initial reversible capacity was 638 mAh g^−1^, which became 436 mAh g^−1^ after 30 cycles. The composite can retain 66% of capacity after 30 cycles.	Low decay in reversible capacity, good cycle ability and capacity retention.	[114]
Amorphous CNT-NiO nanosheet composite	A high discharge capacity of 1034 mAh g^−1^ was delivered after 300 cycles at a relatively 800 mA g^−1^ current density and 98.1% coulombic efficiency	High discharge capacity, coulombic efficiency, and high specific reversible capacity.	[117]
3D G-CNT-Ni nanostructures	It exhibited an initial capacity of 2395.2 mAh g^−1^ with a high reversible capacity of 648.2 mAh g^−1^ after 50 cycles.	High initial capacity, excellent stability, high reversible capacity.	[119]
MWCNT/NiO nanocomposite	Initial discharge and charge capacities of 1083.8 and 720.2 mAh g^−1^, respectively, with 66.45% coulombic efficiency. This sample maintained a stable ~800 mAh g^−1^ discharge capacity and 97% coulombic efficiency after 50 cycles.	High coulombic efficiency, stable discharge capacity, excellent cyclability.	[120]
NiO-G-CNTs nanohybrid composite	The nanohybrids delivered an initial discharge capacity of 1515.1 mAh g^−1^, a stable reversible specific capacity of 1022 mAh g^−1^, also after 50 cycles, a specific capacity of 858.1 mAh g^−1^ at 100 mA g^−1^ current density.	High initial discharge, stable reversible capacity, remarkable cycle stability and rate performance.	[121]
Biochar-CNT-NiO composite	An initial discharge capacity of 981.0 mAh g^−1^ with 65.18% coulombic efficiency.	Highly stable cycle performance, outstanding rate capacity.	[122]
RGO/NiO composite	The initial specific discharge/charge capacities were of 1641 mAh g^−1^ and 1097 mAh g^−1^, respectively, with 1041 mAh g^−1^ specific discharge capacity after 50 cycles at 100 mA g^−1^ current density and an excellent rate capacity of 727 mAh g^−1^ at 1600 mA g^−1^ current density.	High initial specific capacities, highly stable cycle performance and rate capability.	[97]
rGO/NiO nanosheet composite	Specific discharge/charge capacity of 1570 and 1193 mAh g^−1^ with 75.6% coulombic efficiency.	High coulombic efficiency, remarkable rate capability, high cycle stability.	[123]
NiO/rGO composite	It exhibited high reversible capacity of 1036.8 mAh g^−1^, even after 50 cycles.	High reversible capacity, can retain capacity closed to initial capacity.	[125]
3D NiO-Ni nanowire composite	The composite offered a capacity of 827 mAh g^−1^ at 10th cycle, at 100 mA g^−1^ current density. At the 40th cycle the specific capacity reaches 424 mAh g^−1^, at 1000 mA g^−1^ current density, with a stable capacity retention up to 900 mAh g^−1^	Highly stable cyclability, high capacity retention ability.	[132]
3D-flower-like NiO/Ni nanocomposite	The Ni-doped NiO/Ni nanocomposite exhibited discharge/charge of 1316 mAh g^−1^ and 898 mAh g^−1^, respectively.	Stable reversible capacity, better capacity retention, superior cyclability, and rate performance	[137]
NiO-Ni nanocomposite	The first discharge capacity of 1152.4 mAh g^−1^ with 71.2% coulombic efficiency.	Higher reversible capacities and cycling ability	[138]
Spherical NiO/Ni nanocomposite	A high specific capacity of 800 mAh g^−1^ after 50 cycles, at a current density of 0.1C. with a reversible capacity of 450 mAh g^−1^, even at 5C rate, and 635 mAh g^−1^ capacity for 300 cycles at 2C, at 50 °C temperature.	High reversible capacity, excellent rate capability, and significantly long cycle stability.	[139]

## Data Availability

All data is provided in the manuscript.

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
