# Peer review of "A Review on Recent Advancements of Ni-NiO Nanocomposite as an Anode for High-Performance Lithium-Ion Battery"

_nanomaterials, 2022, doi:10.3390/nano12172930_

Round 1
Reviewer 1 Report
This manuscript can be considered for publication if the authors can address the following comments:
1. please improve the quality and double check Figure 1 and Figure 2.
2. as the backgroud in introduction, please update the recent international policy regarding Zero mession.
3, please double check Figure 6a. Nio
4. The overall sentence and grammar need to be checked and corrected. And, most figures have low resolution, which should have better quality. such as Figure 19.
5. suggest add the Funding sources if possible.
6, please double check the reference format. please do more literature review, and add more recent progress and recent published papers
Author Response
Responses to reviewer’s comments
The authors would like to thank the reviewer for the valuable comments on the paper and have made changes in line with the comments.
Reviewer # 1
This manuscript can be considered for publication if the authors can address the following comments:
Comment 1: Please improve the quality and double check Figure 1 and Figure 2.
Response: We appreciate this comment and edited the Figure 1 and Figure 2.
Comment 2: As the background in introduction, please update the recent international policy regarding Zero emission.
Response: We appreciate this comment and addressed this comment in section 1. Introduction by adding following paragraph:
“To alleviate the repercussion of such emissions, the responsible authorities have under-taken recovery policies to decrease emissions to half within 2030 and targeted to attain net-zero emissions by 2050 to accomplish the 1.5 Celsius goal (according to Paris Agreement) [4].”
Comment 3: Please double check Figure 6a. Nio
Response: We rechecked the Figure 6 and changed Nio to NiO.
Comment 4: The overall sentence and grammar need to be checked and corrected. And, most figures have low resolution, which should have better quality. such as Figure 19.
Response: We appreciate this comment and carefully edited the manuscript which is mentioned in the revised manuscript by red color. We tried our best to enhance the Figure 19 and others figures too.
Comment 5: Suggest add the Funding sources if possible.
Response: We do appreciate this comment. However, we mentioned the funding information in page 32 as follows:
“Funding: This work is co-supported by Bangladesh Bureau of Educational Information & Statistics (BANBEIS) through research grant no. PS20191251, Chittagong University of Engineering & Technology (CUET), Bangladesh through research grant no. CUET/REC-06-09/02/2017 and a grant (No. EM220003) from the Korea Institute of Industrial Technology (KITECH). The corresponding author is responsible for ensuring that the descriptions are accurate and agreed by all authors.”
Comment 6: Please double check the reference format. Please do more literature review, and add more recent progress and recent published papers
Response: We have carefully edited the references and kept the format consistent. In addition, we have cited few more research articles during this time of review and editing, which is marked in red color.

Reviewer 2 Report
The author reviews the integration of electrochemical performance of binder involved nanocomposite of NiO as an anode of LIB and epitomizes the synthesis and characterization parameters of nanocomposite anode. However, there are still some questions need to be addressed before considering publication.
1. The abbreviations of Lithium-ion batteries in this paper are not uniform, and “LIB” and “LIBs” are used together. Please carefully check and unify them.
2. The topic of this paper is Ni-NiO nanocomposite, and the author also claims that Ni-NiO nanocomposite is more suitable as anode for LIB compared with other materials. However, there is nothing about Ni-NiO nanocomposite in the introduction. More information on the characteristics of Ni-NiO nanocomposite and the purpose of this review should be added.
3. There are some grammatical and formatting issues in this manuscript. The current version of this manuscript may not be very readable. The author should carefully check the writing.
4. In the third section, “Anode Materials”, the author refers to carbon-based materials as one of the MOs. Is this reasonable?
5. In the paper, the author has been emphasizing nanocomposite and using materials to lead to various anode materials. What is the definition of nanocomposite materials? What kind of material is a nanocomposite? Please give a detailed explanation.
6. In this manuscript, the author only describes the characteristics and properties of the materials reported in each literature one by one, without a general description and unique opinions, which does not seem to be a qualified review. Please sort out the logic and revise it.
7. The format of the references is not uniform, please check carefully.
Author Response
Responses to reviewer’s comments
The authors would like to thank the reviewer for the valuable comments on the paper and have made changes in line with the comments.
Reviewer # 2
The author reviews the integration of electrochemical performance of binder involved nanocomposite of NiO as an anode of LIB and epitomizes the synthesis and characterization parameters of nanocomposite anode. However, there are still some questions need to be addressed before considering publication.
Comment 1: The abbreviations of Lithium-ion batteries in this paper are not uniform, and “LIB” and “LIBs” are used together. Please carefully check and unify them.
Response: We appreciate this comment and carefully edited the manuscript to make this uniform.
Comment 2: The topic of this paper is Ni-NiO nanocomposite, and the author also claims that Ni-NiO nanocomposite is more suitable as anode for LIB compared with other materials. However, there is nothing about Ni-NiO nanocomposite in the introduction. More information on the characteristics of Ni-NiO nanocomposite and the purpose of this review should be added.
Response: We appreciate this comment and edited the manuscript by adding following paragraph in pages 4 and 5:
“Undoubtedly, nanocomposite materials are currently of interest as LIB anode, as they in-corporate nanosized particles within the matrix of standard material possessing high surface area, high surface-to-volume ratio, and electrochemically stable structure. Moreover, nanocomposites demonstrates dramatically improved mechanical strength, structural stability, toughness, and thermal or electrical conductivity. These characteristics pave the way for new reaction sites, shorten Li+ transportation distance, improve the kinetics of Li+, and enhance specific capacity and cycle stability, which makes them feasible anode material. It is noted that the nanocomposites of NiO have grabbed particular attention as LIB anode whereas the Ni-NiO nanocomposite is the most promising one. In this review article, we tried to elucidate the issues regarding conventional anodes, and how to recuperate the drawbacks by introducing and analyzing the performance of different materials as anodes. Moreover, we briefly discussed different nanocomposite materials and particularly highlighted on binder involved Ni-NiO nanocomposite materials as LIB anode. Furthermore, the advantages and disadvantages of Ni-NiO nanocomposite are elucidated.
Hence, this review is summarized as follows: Firstly, we provided the insertion/extraction mechanism of LIB in short. Subsequently, we demonstrated a brief over-view of different anode materials based on reaction mechanisms and their nanocomposites including major challenges. Thereafter, we focused on the performance of NiO nanocomposites with carbon, graphene nanosheets, carbon nanotubes, reduced graphene oxides, and how Ni-NiO nanocomposite outweighs the disadvantages of the abovementioned nanocomposites. Finally, we concluded this review with some noteworthy concluding remarks and future research directions.”
Comment 3: There are some grammatical and formatting issues in this manuscript. The current version of this manuscript may not be very readable. The author should carefully check the writing.
Response: We appreciate this comment and edited the manuscript accordingly, which is mentioned by red color text in revised manuscript.
Comment 4: In the third section, “Anode Materials”, the author refers to carbon-based materials as one of the MOs. Is this reasonable?
Response: This is not reasonable. We apologies for this misinformation and deleted this sentence in revised manuscript.
Comment 5: In the paper, the author has been emphasizing nanocomposite and using materials to lead to various anode materials. What is the definition of nanocomposite materials? What kind of material is a nanocomposite? Please give a detailed explanation.
Response: We appreciate these comments and addressed these comments in page 7 by adding the following paragraph:
“Currently, nanocomposite materials are drawing extensive research interest as anode for LIB. In general, nanocomposites are multiphasic materials in which a phase must possess one, two, or three dimensions below 100 nm, or nanoscale distances exist between the phases. Nanocomposites can be formed through amalgamating inorganic nanoclusters, metals, oxides, and semiconductors with different metallic compounds. Nanocomposites have emerged as favorable alternatives to bulk engineering materials that have certain limitations. The presence of nanoparticles phase within the composite structure facilitates catalytic activity and substantial improvements in mechanical properties, flexi-bility, thermal stability, and improved electrical conductivity [38,39]. It is noted that some nanocomposite materials are 1000 times tougher as compared to their bulk counterparts. These unique characteristics of nanocomposites mainly arise from small-sized particles, high surface area, and possible interfacial interaction between constituent phases. Consequently, these promising characteristics lead nanocomposites to be employed as propi-tious anode material for LIB.”
Comment 6: In this manuscript, the author only describes the characteristics and properties of the materials reported in each literature one by one, without a general description and unique opinions, which does not seem to be a qualified review. Please sort out the logic and revise it.
Response: We do appreciate this comment. However, the scope this review paper is to comprehensive analysis of electrochemical performances of Ni-NiO composites as anode of LIBs.
Comment 7: The format of the references is not uniform, please check carefully.
Response: We appreciate this comment and edited the references.

Reviewer 3 Report
This review is aimed at highlighting the latest results in the study of anode materials based on nickel oxide nanostructures and their application in the field of lithium-ion batteries. The presented topic is very interesting and informative, and the data collected by the authors fully cover the latest achievements in the field of synthesis and characterization of nanostructured materials based on nickel oxide, which have great prospects in this direction. In general, the review is quite well written, and also covers all aspects related to the selected objects of study and their applicability as anode materials, however, before accepting it for publication, the authors should consider the following issues in more detail.
1. In the abstract, the authors should give a more detailed description of the expediency of replacing classical carbon materials with nickel oxide, as well as its differences from other types of oxides used in this direction.
2. The authors should briefly indicate the prospects for the use of various types of ionizing radiation as one of the methods for modifying nickel nanostructures used as anode materials.
3. The results presented in Table 2, if possible, can be supplemented with images of the obtained nanostructures, so that it is more clear about the effect of morphology on performance.
Author Response
Responses to reviewer’s comments
The authors would like to thank the reviewer for the valuable comments on the paper and have made changes in line with the comments.
Reviewer # 3
This review is aimed at highlighting the latest results in the study of anode materials based on nickel oxide nanostructures and their application in the field of lithium-ion batteries. The presented topic is very interesting and informative, and the data collected by the authors fully cover the latest achievements in the field of synthesis and characterization of nanostructured materials based on nickel oxide, which have great prospects in this direction. In general, the review is quite well written, and also covers all aspects related to the selected objects of study and their applicability as anode materials, however, before accepting it for publication, the authors should consider the following issues in more detail.
Comment 1: In the abstract, the authors should give a more detailed description of the expediency of replacing classical carbon materials with nickel oxide, as well as its differences from other types of oxides used in this direction.
Response: We appreciate this comment. However, we have detailed explained why NiO would be the competitive choices as anode of LIBs instead of classical carbon materials in the manuscript (pages 4 & 5)). Hence, we omitted this explanation in Abstract.
“Nevertheless, long-range electric vehicles experience inadequate power density, as 150-265 Wh/kg energy density is not sufficient enough[17]. Additionally, the limited capacity of graphite anode (372 mAh g-1) is inadequate for long-run EVs, HEVs, and other portable devices as they require high energy density as well as high power density [18,19]. Moreover, significant structural collapse and exfoliation during cycling, strong polarization at a fast charge/discharge current rate, formation of lithium dendrites on the graphite surface, low working voltage (0.1V vs. Li/Li+), low working temperature, heating of the device are the major concerns [20]. Furthermore, the conventional electrode of commercial LIB was constructed through slurry coating procedure where bulk carbon,Super P and PVDF amalgamated together and coated onto current collector. During the electrochemical operation, the extensive volume change in the active material causes the material to deflect from the current collector. This distorted active material falls into and mixes with the electrolyte as dead weight thus diminishing the capacity of the cell. Consequently, there is an urgency for designing and developing suitable anode material, an improvement in existing materials, and substituting graphite which can hold plenty of lithium-ion within. These may improve the structural stability, and the capacity of the battery for the applications like electrification of vehicles, and grid-scale energy storage. Undoubtedly, nanocomposite materials are currently of interest as LIB anode, as they incorporate nanosized particles within the matrix of standard material possessing high surface area, high surface-to-volume ratio, and electrochemically stable structure. Moreover, nanocomposites demonstrates dramatically improved mechanical strength, structural stability, toughness, and thermal or electrical conductivity. These characteristics pave the way for new reaction sites, shorten Li+ transportation distance, improve the kinetics of Li+, and enhance specific capacity and cycle stability, which makes them feasible anode material. It is noted that the nanocomposites of NiO have grabbed particular attention as LIB anode whereas the Ni-NiO nanocomposite is the most promising one. In this review article, we tried to elucidate the issues regarding conventional anodes, and how to recuperate the drawbacks by introducing and analyzing the performance of different materials as anodes. Moreover, we briefly discussed different nanocomposite materials and particularly highlighted on binder involved Ni-NiO nanocomposite materials as LIB anode. Furthermore, the advantages and disadvantages of Ni-NiO nanocomposite are elucidated.
Hence, this review is summarized as follows: Firstly, we provided the insertion/extraction mechanism of LIB in short. Subsequently, we demonstrated a brief overview of different anode materials based on reaction mechanisms and their nanocomposites including major challenges. Thereafter, we focused on the performance of NiO nanocomposites with carbon, graphene nanosheets, carbon nanotubes, reduced graphene oxides, and how Ni-NiO nanocomposite outweighs the disadvantages of the abovementioned nanocomposites. Finally, we concluded this review with some noteworthy concluding remarks and future research directions.”
Comment 2: The authors should briefly indicate the prospects for the use of various types of ionizing radiation as one of the methods for modifying nickel nanostructures used as anode materials.
Response: We do appreciate this comment and addressed this comment by adding following paragraph in pages 23 and 24:
“Kozlovskiy et al [133] reported the ionizing radiation method to enhance the efficiency of performance of Ni-NiO nanotubes as anodes of LIBs. The Ni-NiO nanotube structured anode exhibited stable discharge capacity of 900 mAh-1 after three charge/discharge cycles. Furthermore, Kozlovskiy et al [134] reported the increase in Ni nanotubes lifetime when used as anode of LIBs. It is noted that modified nanostructures exhibited an increase in the number of cycles from 344 to 607 and 651. The increase in the lifetime of the anode material for irradiated samples was due to the removal of stresses and distortions in the crystal lattice, as well as partial annealing of defects as a result of irradiation [135–137].”
Comment 3: The results presented in Table 2, if possible, can be supplemented with images of the obtained nanostructures, so that it is more clear about the effect of morphology on performance.
Response: We do appreciate this comment. However, we reported most of the images as Figures in the manuscript before presenting Table 2. Hence, we have carefully avoided the repeated data in the manuscript.

Round 2
Reviewer 1 Report
as for comments 1, please double check Figure 2. the reviewer mean please check the energy density value.
Author Response
Responses to reviewer’s comments
The authors would like to thank the reviewer for the valuable comments on the paper and have made changes in line with the comments.
Reviewer # 1 R2
Comment 1: English language and style are fine/minor spell check required.
Response: We appreciate this comment. We have carefully edited the English language and change is made in red text in the revised manuscript.
Comment 2: Please double check Figure 2. The reviewer mean please check the energy density value.
Response: We double checked the Figure 2 and edited accordingly as follows:
Figure 2. Comparison between LIBs and other batteries in terms of energy densities. Reproduced with permission from [9]
